# p27[Kip1] promotes invadopodia turnover and invasion through the regulation of the PAK1/Cortactin pathway

Pauline Jeannot[1,2,3], Ada Nowosad[1,2,3], Renaud T Perchey[1,2,3], Caroline Callot[1,2,3], Evangeline Bennana[4], Takanori Katsube[5], Patrick Mayeux[4], François Guillonneau[4], Stéphane Manenti[1,2,3], Arnaud Besson[1,2,3]*

[1]INSERM UMR1037, Cancer Research Center of Toulouse, Toulouse, France; [2]Université Toulouse III Paul Sabatier, Toulouse, France; [3]CNRS ERL5294, Toulouse, France; [4]3P5 proteomics facility of the Université Paris Descartes, Inserm U1016 Institut Cochin, Sorbonne Paris Cité, Paris, France; [5]National Institute of Radiological Sciences, Chiba, Japan

**Abstract** p27[Kip1] (p27) is a cyclin-CDK inhibitor and negative regulator of cell proliferation. p27 also controls other cellular processes including migration and cytoplasmic p27 can act as an oncogene. Furthermore, cytoplasmic p27 promotes invasion and metastasis, in part by promoting epithelial to mesenchymal transition. Herein, we find that p27 promotes cell invasion by binding to and regulating the activity of Cortactin, a critical regulator of invadopodia formation. p27 localizes to invadopodia and limits their number and activity. p27 promotes the interaction of Cortactin with PAK1. In turn, PAK1 promotes invadopodia turnover by phosphorylating Cortactin, and expression of Cortactin mutants for PAK-targeted sites abolishes p27's effect on invadopodia dynamics. Thus, in absence of p27, cells exhibit increased invadopodia stability due to impaired PAK1-Cortactin interaction, but their invasive capacity is reduced compared to wild-type cells. Overall, we find that p27 directly promotes cell invasion by facilitating invadopodia turnover via the Rac1/PAK1/Cortactin pathway.

*For correspondence: arnaud. besson@inserm.fr

## Introduction

p27[Kip1] is a cell cycle inhibitor that binds to a broad range of cyclin-CDK complexes (*Besson et al., 2008*). p27-mediated cyclin-CDK inhibition involves the N-terminal extremity (aa 28–89) of the protein that allows interaction with both cyclin and CDK subunits (*Ou et al., 2012*; *Besson et al., 2008*). The critical role of p27 as a negative regulator of cell proliferation is underscored by the phenotype of p27 knockout mice, which exhibit a hyperproliferative phenotype in multiple tissues and are more susceptible to tumor development than wild-type animals (*Fero et al., 1996*; *Kiyokawa et al., 1996*; *Nakayama et al., 1996*; *Besson et al., 2006*; *Fero et al., 1998*). While the tumor suppressive role of p27 is confirmed by the frequent loss of p27 expression in many types of cancers, p27 is also mislocalized in a fraction of tumors due to activation of various signaling pathways that cause its nuclear export and/or cytoplasmic retention (*Chu et al., 2008*). Cytoplasmic p27 has been associated with decreased patient survival, high tumor grade and metastasis in several tumor types including breast carcinomas, acute myeloid leukemia, glioblastoma, melanoma and non-small cell lung carcinomas (*Lin et al., 2016*; *Chu et al., 2008*; *Liang et al., 2002*; *Yang et al., 2011*; *Min et al., 2004*; *Cheng et al., 2015*). The possibility that cytoplasmic p27 may confer a pro-tumorigenic advantage was confirmed in a mouse model expressing a mutant form of p27 that cannot bind cyclin-CDK complexes (p27CK-) (*Besson et al., 2007*; *Serres et al., 2011*). These mice are more

**eLife digest** When animals develop from embryos to adults, or try to heal wounds later in life, their cells have to move. Moving means that the cells must invade into their surroundings, a dense network of proteins called the extracellular matrix. The cell first attaches to the extracellular matrix; degrades it; and then moves into the newly opened space. Cells have developed specialized structures called invadosomes to enable all these steps. Invadosomes are never static, they first assemble where cells interact with extracellular matrix, they then release proteins that loosen the matrix, and finally disassemble again to allow cells to move. Invadosomes in cancer cells often become overactive, and can allow the tumor cells to spread throughout the body.

A lot of different proteins are involved in controlling how and when cells move. p27 is a well-known protein usually found in a cell's nucleus along with the cell's DNA. Inside the nucleus, p27 suppresses tumor growth by stopping cells from dividing. However, often in cancer cells p27 moves outside of the cell's nucleus where it contributes to cell movement via an unknown mechanism.

To answer how p27 controls cell invasion, Jeannot et al. used a biochemical technique to uncover which proteins p27 binds to when it is outside of the nucleus. One of its interaction partners was called Cortactin. This protein is known to be an important building block of invadosomes, and is involved in both the assembly and disassembly of these structures. In further experiments, Jeannot studied mouse cells with or without p27 and human cancer cells that can be grown in the laboratory. The experiments revealed that p27 promotes an enzyme called PAK1 to also bind to Cortactin. PAK1 then modified Cortactin, causing whole invadosomes to disassemble, which in turn allowed cells to de-attach from the matrix and move forward. In contrast, cells lacking p27 had more stable invadosomes, attached more strongly to the matrix and were better at degrading it, but could not invade as well as cells with p27.

Overall these experiments showed a new way that p27 promotes cell invasion. The next steps will include finding out exactly how the modification of Cortactin causes the invadosomes to disassemble. Furthermore, it will be important to study whether forcing p27 back into the nucleus can reduce the spread of cancer cells in the body.

susceptible than p27-null and wild-type mice to both spontaneous and induced tumor development, thus uncovering an oncogenic role for p27 in vivo (*Serres et al., 2011*; *Besson et al., 2007*).

It now appears that p27 is a multifunctional protein involved in the regulation of multiple cellular processes, some of which are regulated by p27 in a CDK-independent manner (*Besson et al., 2008*; *Sharma and Pledger, 2016*). Indeed, p27 has been implicated in the control of cell migration, transcriptional repression, autophagy, stem cell specification and differentiation, cytokinesis, and apoptosis (*Besson et al., 2008*; *Sharma and Pledger, 2016*; *Baldassarre et al., 2005*; *Besson et al., 2004b*, *2007*; *Pippa et al., 2012*; *Serres et al., 2012*; *Nickeleit et al., 2008*; *Liang et al., 2007*; *Li et al., 2012*; *Nguyen et al., 2006*; *Jeannot et al., 2015*). This functional versatility may stem from the intrinsically disordered structure of p27, which folds upon binding to other proteins, allowing p27 to interact with a wide variety of partners (*Lacy et al., 2004*; *Galea et al., 2008*). While these various functions participate in homeostasis in normal cells and tissues, they may be co-opted by tumor cells to promote oncogenesis.

Metastasis, the process whereby cells from a primary tumor disseminate throughout the body and form secondary tumors at distant sites, is the leading cause of cancer related death (*Sethi and Kang, 2011*). It requires tumor cells to migrate and invade through neighboring tissues, enter the blood flow and then extravasate to disseminate and form new tumors in host tissues (*Sethi and Kang, 2011*). Regulation of cell motility was the first CDK-independent role ascribed to p27 (*Nagahara et al., 1998*; *Denicourt et al., 2007*; *Besson et al., 2004b*). In the cytosol, p27 can interact with RhoA, preventing RhoA activation by guanine-nucleotide exchange factors (GEFs) and thereby modulating actin cytoskeleton dynamics and migration (*Besson et al., 2004a*, *2004b*). Consequently, mouse embryo fibroblasts (MEFs) lacking p27 have more RhoA activity, increased numbers of stress fibers and focal adhesions and exhibit a defect in migration (*Besson et al., 2004b*). This pathway is important for proper migration of bone marrow

macrophages and developing cortical neurons and for cancer cell migration and invasion in vivo (*Godin et al., 2012*; *Nguyen et al., 2006*; *Papakonstanti et al., 2007*; *Gui et al., 2014*; *Wu et al., 2006*; *See et al., 2010*; *Larrea et al., 2009*; *Jin et al., 2013*). p27 can also regulate cell migration by controlling microtubule stability through Stathmin or directly by binding to microtubules and promoting microtubule polymerization (*Baldassarre et al., 2005*; *Godin et al., 2012*). In a 3D environment, cells adopt different strategies to migrate and invade through surrounding matrix (*Petrie and Yamada, 2016*; *Sethi and Kang, 2011*). The mode of migration is influenced, at least in part, by the activities of Rho GTPases: RhoA activity favors an amoeboid migration, whereas Rac1 activity promotes mesenchymal migration (*Sahai and Marshall, 2003*; *Vial et al., 2003*). Accordingly, cytoplasmic p27 promotes mesenchymal migration via the inhibition of RhoA activity, while cells lacking p27 tend to adopt an amoeboid mode of migration (*Gui et al., 2014*; *Belletti et al., 2010*). More recently, p27 was reported to promote epithelial to mesenchymal transition by binding to JAK2, promoting STAT3 activation and the upregulation of Twist1 (*Zhao et al., 2015*).

Here, we describe a novel mechanism by which p27 directly regulates invadopodia formation and cell invasion through extracellular matrix (ECM) via Cortactin. Invadosomes (designating both invadopodia and podosomes) are thought to allow cells to coordinate ECM degradation with migration within the tissue microenvironment (*Murphy and Courtneidge, 2011*; *Linder et al., 2011*; *Di Martino et al., 2016*). Cortactin plays a key role in the formation of actin protrusive structures such as lamellipodia and invadosomes (*Murphy and Courtneidge, 2011*; *Kirkbride et al., 2011*; *MacGrath and Koleske, 2012*). Cortactin has been involved in all steps of the invadosome lifecycle, from assembly, maturation, proteolytic activity and disassembly (*MacGrath and Koleske, 2012*; *Murphy and Courtneidge, 2011*; *Moshfegh et al., 2014*). All these steps appear to be regulated by phosphorylation events on Cortactin (*Oser et al., 2009*; *Moshfegh et al., 2014*; *Murphy and Courtneidge, 2011*). Overall, Cortactin is a scaffold protein composed of an N-terminal acidic domain, followed by several actin-binding repeats allowing its interaction with F-actin, a Pro-rich region and a SH3 domain (*MacGrath and Koleske, 2012*; *Kirkbride et al., 2011*). Phosphorylation of Cortactin by Src and Abl family tyrosine kinases has two effects: first, this releases the actin severing protein Cofilin from an inhibitory interaction with Cortactin and generates actin barbed ends; second, Arp2/3, N-WASP and Nck1 are recruited onto Cortactin, promoting the polymerization of branched actin (*Oser et al., 2009*; *Weaver et al., 2002*; *Oser et al., 2010*). Dephosphorylation of Cortactin then allows the inhibition of Cofilin activity and stabilizes the invadopodia (*Oser et al., 2009*). Cortactin also plays a key role in matrix metalloproteinase secretion in mature invadopodia (*Clark et al., 2007*). Finally, invadopodia disassembly is induced by sequential activation of the Rac-GEF Trio, Rac1 and p21-Activated Kinase-1 (PAK1) pathway and presumably by PAK1-mediated phosphorylation of Cortactin on Ser113, since a Ser113 to Ala mutant blocked invadopodia disassembly (*Moshfegh et al., 2014*). How PAK-phosphorylated Cortactin mediates invadopodia disassembly is still unclear but could be due, at least in part, to a decreased affinity of Cortactin phosphorylated within its F-actin binding repeats (on S113, S150 and/or S282) for F-actin, potentially destabilizing the structure (*Webb et al., 2006*, *2005*).

We found that p27 binds to Cortactin and localizes to invadopodia following serum or growth factor stimulation and promotes cell invasion. Paradoxically, p27-null cells have more invadopodia and degrade gelatin more efficiently, but exhibit impaired invasive capacity. In fact, we found that p27 promotes the recruitment of PAK1 to Cortactin and invadopodia turnover and this is dependent on phosphorylation of S113/S150/S282 of Cortactin, the sites targeted by PAK kinases. Thus, in absence of p27, the dynamics of invadopodia is altered and prevents efficient invasion through ECM. Altogether, we have identified a novel mechanism by which p27 directly controls cell invasion that could contribute to the increased invasive and metastatic capacity of tumors where p27 is mislocalized in the cytoplasm.

## Results

### p27 interacts with Cortactin and localizes to invadopodia

While the role of p27 in the regulation of cell migration is firmly established, how p27 is targeted to specific locations in the cytoplasm to control motility and invasion remains unclear (*Gui et al., 2014*; *Belletti et al., 2010*; *Besson et al., 2004a*, *2004b*; *Godin et al., 2012*; *Nguyen et al., 2006*;

*Papakonstanti et al., 2007*; *Wu et al., 2006*; *See et al., 2010*; *Larrea et al., 2009*; *Jin et al., 2013*). We recently identified Cortactin in a proteomic screen in which protein arrays (Protoarray, Thermo-Fisher Scientific) were probed with recombinant human p27, indicating that the two proteins interact directly. We confirmed the binding of p27 to Cortactin in HEK 293 cells overexpressing p27 and Myc-tagged Cortactin (*Figure 1A*), as well as on endogenous proteins in Hela cells and MEFs immortalized with the human papilloma virus E6 protein (*Figure 1B*) (*Serres et al., 2011*). The interaction domain of Cortactin on p27 was mapped by pull-down assays using various GST-p27 fusion proteins and Myc-Cortactin in HEK 293 cells. Cortactin bound to the p27CK- mutant, that cannot interact with cyclins and CDKs (*Besson et al., 2006, 2007*), and to the C-terminal half (aa 88–198) of p27 but not the N-terminal half (aa 1–87) of the protein that contains the cyclin and CDK interaction domains (*Figure 1C and D*). The Cortactin interaction domain on p27 was narrowed down to the last 8 aa of p27, as a p27CK- 1–190 mutant did not interact with Cortactin any longer (*Figure 1E*). A p27CK- 1–197 mutant, lacking only the C-terminal threonine residue, still bound to Cortactin (*Figure 1E*), suggesting that phosphorylation on T198 is not needed for binding to Cortactin.

Similarly, pull-down assays using full-length GST-p27 were performed on HEK 293 cell lysates expressing various Myc-tagged Cortactin truncation mutants to map the p27 interaction domain on Cortactin (*Katsube et al., 2004*). These experiments revealed that p27 interacts with the N-terminal half of Cortactin, within the actin binding repeats, as a mutant Cortactin lacking the actin binding repeats 3 to 6 (ΔABR3-6) did not bind p27 (*Figure 1F*).

Since Cortactin is a key component of invadopodia, we determined if p27 could colocalize with Cortactin in these structures. To remove most of soluble p27 prior to fixation and immunostaining, p27+/+E6 MEFs were permeabilized with digitonin, as described previously (*Serres et al., 2012*). In these conditions, p27 could be readily observed in invadopodia where it colocalized with Tks5, a commonly used invadopodia marker (*Seals et al., 2005*), or Cortactin (*Figure 2A and B*, respectively). The same approach was used to confirm p27 localization to invadopodia in human A549 lung adenocarcinoma and A375 melanoma cell lines (*Figure 2—figure supplement 1*).

Invadopodium and podosome formation is stimulated by growth factors (*Murphy and Court-neidge, 2011*). Growth factor stimulation causes the degradation of p27 as cells enter the cell cycle but also the relocalization of a fraction of p27 in the cytoplasm between 1 hr and 6 hr post-stimulation, dependent on the phoshorylation of p27 on Ser10 (*Besson et al., 2006, 2004b*; *Boehm et al., 2002*; *Ishida et al., 2002*; *Rodier et al., 2001*; *Connor et al., 2003*). Consistent with the translocation of p27 in the cytoplasm and the formation of invadopodia after serum or growth factor stimulation, an increased association between p27 and Cortactin by co-immunoprecipitation at 1 hr and 3 hr post-stimulation in p27+/+E6 MEFs (*Figure 2C*) and in Hela cells (*Figure 2—figure supplement 2*) was observed in five and three independent experiments, respectively. However, due to variability in signal intensities among independent experiments, these differences were not statistically significant. Together, our data indicate that a fraction of cytoplasmic p27 localizes to invadopodia after growth factor stimulation, where it binds to Cortactin.

## p27 regulates invadopodia formation and promotes invasion

Invadopodia and podosomes are capable of both adhering to and degrading ECM (*Murphy and Courtneidge, 2011*; *Di Martino et al., 2016*). Fibroblasts do not normally form invadosomes unless transformed by Src (*Murphy and Courtneidge, 2011*). We first verified that the E6 immortalized MEF lines used in our study were forming functional, matrix degrading invadopodia. For this, p27+/+, p27−/− and p27CK-/CK- E6 MEFs were seeded on fluorescent gelatin and Tks5 immunostaining indicated that these cells could all form Tks5-containing structures that efficiently degraded gelatin (*Figure 3—figure supplement 1*). p27 colocalized with Tks5 at sites of gelatin degradation in MEFs and A549 cells, suggesting that p27 can be present at functional invadopodia (*Figure 3—figure supplement 2*). Since p27 binds to Cortactin and can localize to invadopodia, we determined if p27 status influenced the ability of cells to form invadopodia. Counting of the number of cells forming invadopodia using Tks5 immunostaining (*Figure 3A*) revealed that while cells expressing p27 or p27CK- rarely formed invadopodia (5.83% and 8.68%, respectively), nearly half of MEFs lacking p27 had invadopodia (44.66%) (*Figure 3B*). In keeping with the number of invadopodia forming cells, measurement of the area of fluorescent gelatin degraded per cell showed that p27−/− cells had a dramatically increased capacity to degrade ECM compared to p27+/+ and p27CK-/CK- MEFs (*Figure 3C and D*).

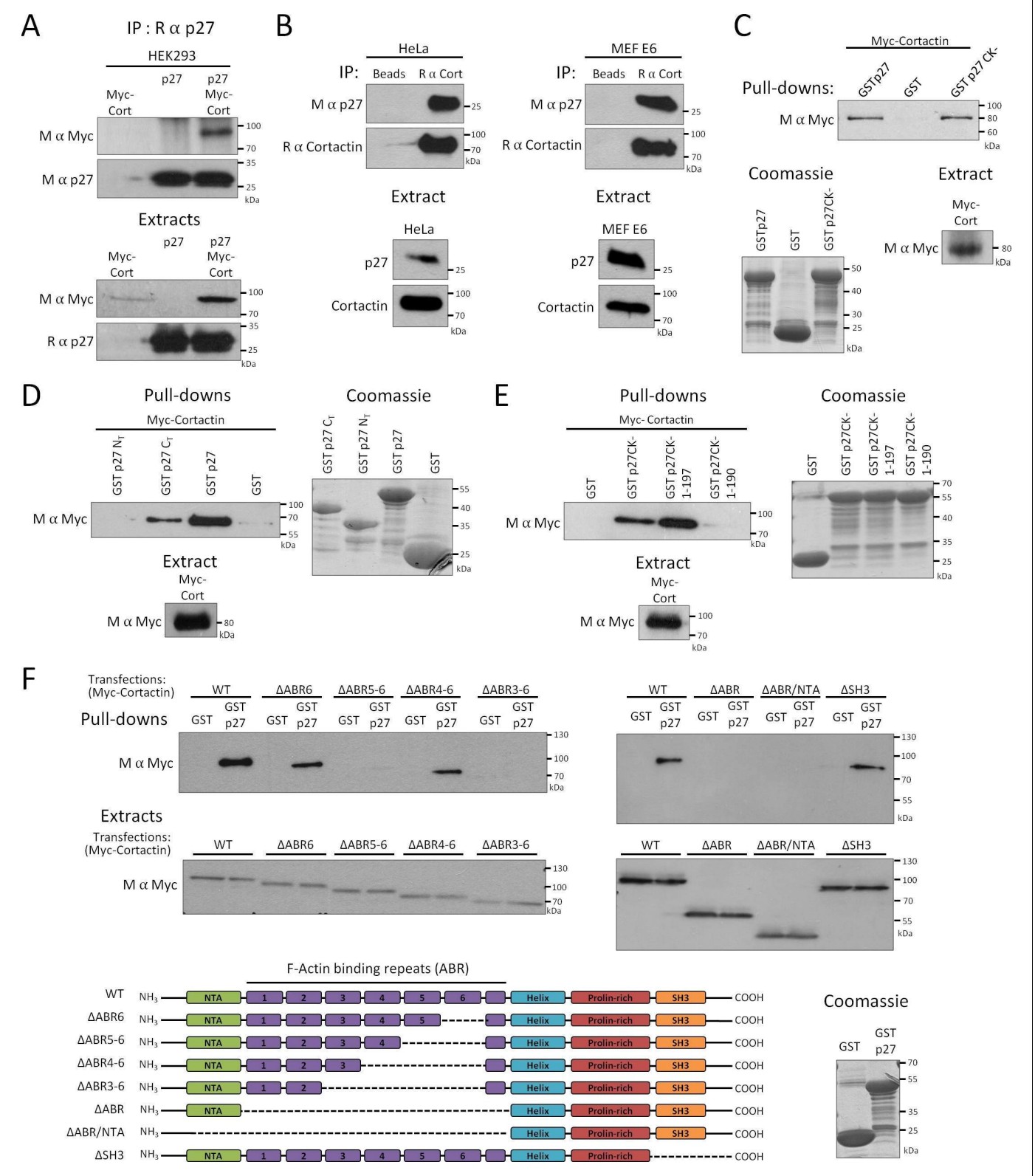

**Figure 1.** p27 binds to Cortactin. (**A–B**) Co-immunoprecipitation of p27 and Cortactin: (**A**) p27 was immunoprecipitated using rabbit anti-p27 (C19) antibodies from HEK293 lysates transfected with plasmids encoding p27, Myc-Cortactin (Myc-Cort) or both. (**B**) Immunoprecipitation of endogenous Cortactin using rabbit anti-Cortactin (H191) antibodies from HeLa or E6 MEF lysates, beads alone were used as control. (**A–B**) Co-immunoprecipitated proteins were detected by immunoblot with mouse anti-c-Myc (9E10) (**A**) or mouse anti-p27 (F-8) antibodies (**B**). Immunoprecipitated proteins were

*Figure 1 continued on next page*

*Figure 1 continued*

visualized by reprobing the membrane with mouse anti-p27 (F-8) (**A**) or with rabbit anti-Cortactin (H-191) antibodies and anti-rabbit Ig light-chain secondary antibodies (**B**). Immunoblots of extracts show the level of proteins in each condition. (**C–F**) Pull-down assays: HEK293 cells were transfected with Myc-Cortactin (**C–E**) or various deletion mutants of Myc-Cortactin (**F**) (ΔABR6, ΔABR5-6, ΔABR4-6, ΔABR3-6, ΔABR, ΔABR/NTA or ΔSH3, described in the schematic representation of Cortactin, bottom panel). NTA: N-Terminal acidic domain; ABR: actin binding repeat; Helix: helical domain; SH3: Src-homology three domain. Lysates were subjected to pull-down assays using GST, GST-p27 or GST-p27CK- (**C**), or GST, GST-p27, GST-p27 $N_T$ (1-87) and GST-p27 $C_T$ (88-198) (**D**), or GST, GST-p27CK-, GST-p27CK- (1-197) and GST-p27CK- (1-190) (**E**) or GST and GST-p27 (**F**). The amounts of Myc-Cortactin bound to the beads and of transfected protein present in the extracts were detected by immunoblot using mouse anti c-Myc (9E10) antibodies. The amounts of GST or GST p27 and mutants used in the assays were visualized by Coomassie staining of the gels. (**A–F**) All panels show representative results of at least three independent experiments.

To confirm the involvement of p27 in regulating invadopodia number and proteolytic activity, we retrovirally infected p27−/− E6 MEFs with either empty vector, wild-type (WT) p27, p27CK- or a p27CK- 1–190 truncation mutant (*Besson et al., 2004a*; *Serres et al., 2012*) that cannot interact with Cortactin (*Figure 1E*). Expression levels of p27 in these infected cells were determined by immunoblot (*Figure 3E*). Quantification of the number of cells forming invadopodia and of the area of gelatin degraded per cell indicated that expression of WT p27 and p27CK- significantly decreased both the number of p27−/− cells forming invadopodia and their capacity to degrade gelatin, while the p27CK- 1–190 mutant had no effect (*Figure 3F and G*). Thus, our data suggests that p27 limits invadopodia formation and that the domain mediating its interaction with Cortactin is required for this function.

The finding that p27 knockout cells more frequently form invadopodia and exhibit increased ECM degradation activity was surprising given the previous reports that p27 promotes migration and invasion (*Denicourt et al., 2007*; *Godin et al., 2012*; *Nguyen et al., 2006*; *Papakonstanti et al., 2007*; *Gui et al., 2014*; *Wu et al., 2006*; *See et al., 2010*; *Larrea et al., 2009*; *Jin et al., 2013*; *Zhao et al., 2015*; *Besson et al., 2004b*). We compared the motility of p27+/+, p27−/− and p27CK-/CK- E6 MEFs in 2D scratch wound assays and p27−/− cells had a migration defect (*Figure 4A and B*), in agreement with our previous findings (*Besson et al., 2004b*). We next measured the capacity of these MEFs to invade through a layer of Collagen I in transwell inserts. Similar to 2D migration, 3D invasion was reduced in p27−/− MEFs compared to either WT p27 or p27CK-expressing cells (*Figure 4C and D*), consistent with previous reports (*Denicourt et al., 2007*; *Gui et al., 2014*; *Wu et al., 2006*; *See et al., 2010*; *Jin et al., 2013*; *Zhao et al., 2015*). To confirm the involvement of p27 in regulating invasion, p27−/− E6 MEFs retrovirally infected with either empty vector, p27CK- or p27CK- 1–190 (*Figure 4E*, left panel) were subjected to transwell invasion assays. While p27CK- restored invasion of p27−/− cells, the p27CK- 1–190 mutant had no effect (*Figure 4E*, right panel). Taken together our results indicate that although p27 limits the number of invadopodia in MEFs, it also promotes cellular invasion through ECM. On the other hand, while p27-null cells more frequently form invadopodia and exhibit an increased capacity to degrade ECM, their ability to invade through matrix was significantly impaired.

## p27 promotes the interaction of Cortactin with PAK1

Interestingly, a similar, seemingly paradoxical finding was recently reported, in which inhibition of Rac1, Trio or PAK1 with siRNAs caused a dramatic increase in invadopodia lifetime and in their capacity to degrade matrix, but this was accompanied by a sharp decrease in cell invasion (*Moshfegh et al., 2014*). In this study, activation of the Trio/Rac1/PAK1 pathway induced a putative PAK1-mediated phosphorylation of Cortactin and invadopodia disassembly, as expression of a S113A Cortactin mutant blocked this pathway (*Moshfegh et al., 2014*). In this study, altering the dynamics of invadopodia turnover inhibited invasion (*Moshfegh et al., 2014*).

To find out whether p27-null cells more frequently form invadopodia than p27+/+ cells (*Figure 3A and B*) due to an increase in their lifetime, we infected p27+/+ and p27−/− MEFs with Tks5-GFP to visualize invadopodia in live cells. Videomicroscopy analyses revealed that while invadopodia lifetime was on average 16.7 min in p27+/+ cells, their mean duration was 65.9 min in p27−/− cells (*Figure 5A*). Thus, invadopodia appear more stable in cells lacking p27.

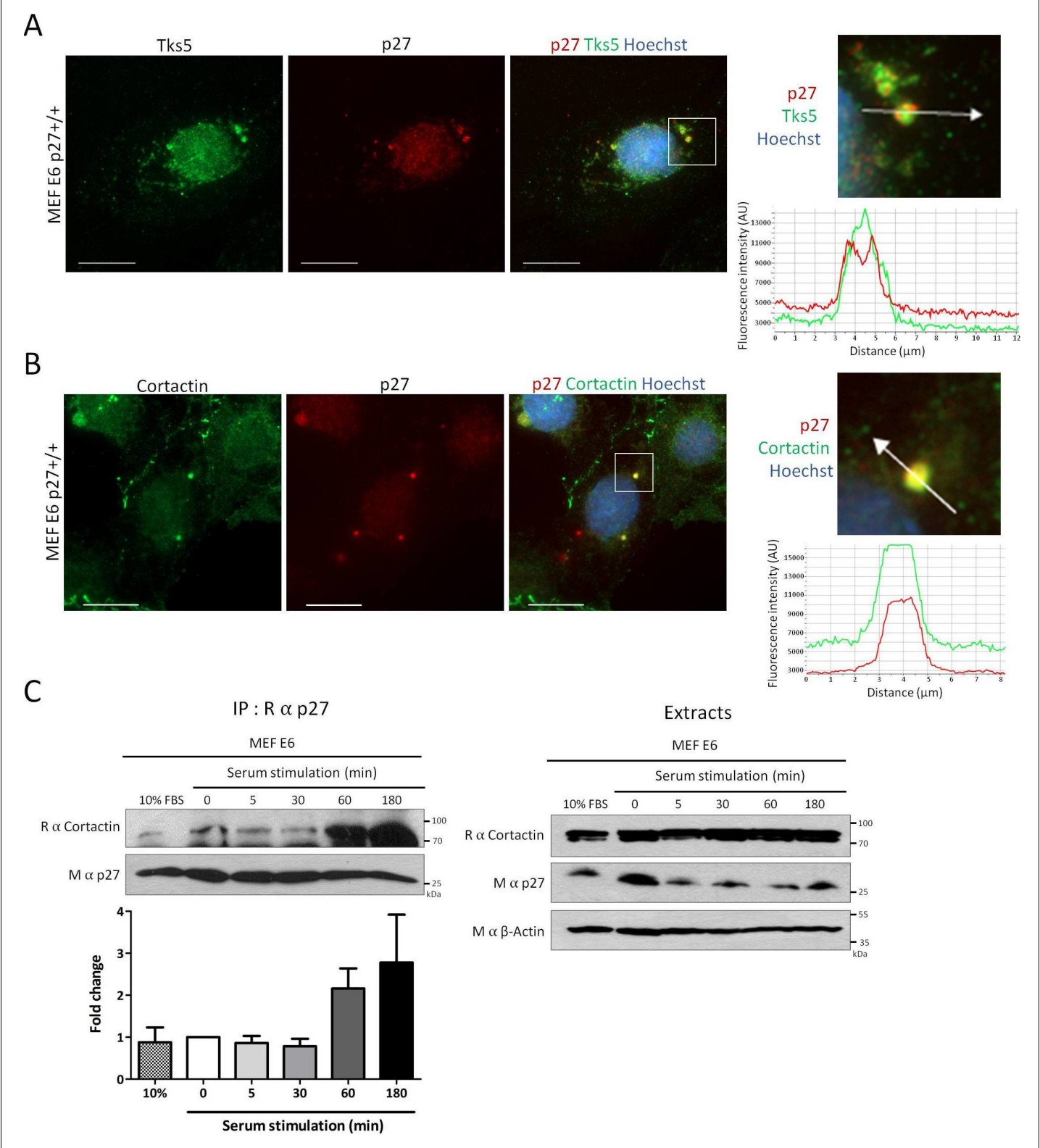

**Figure 2.** p27 colocalizes with Cortactin in invadopodia. (**A–B**) p27+/+ MEFs were seeded on gelatin-coated coverslips for 24 hr. Cells were permeabilized with digitonin prior to fixation. Cells were labeled with mouse anti p27 (SX53G8.5) in red (**A–B**) and rabbit anti-Tks5 (M-300) (**A**) or rabbit anti-Cortactin (H-191) (**B**) in green. Images were acquired using a 60x objective and images displayed are cropped areas. Graphs displaying the fluorescence intensity (arbitrary unit) under the arrows in the enlarged panels were generated with the NIS Element software. Scale bars: 20 μm. (**C**) p27

*Figure 2 continued on next page*

*Figure 2 continued*

+/+ E6 MEFs were starved overnight in DMEM-0.1% FCS and then stimulated with growth medium for the indicated times. Cell lysates were subjected to immunoprecipitation using rabbit anti-p27 (C–19). Immunoprecipitated proteins and corresponding cell extracts were immunoblotted with rabbit anti-Cortactin (H–191) and mouse anti-p27 (F–8) antibodies. β-actin was used as loading control. The graph represents the mean fold change in amounts of Cortactin co-precipitated with p27 in each condition compared to time zero from five independent experiments. These differences were not statistically significant.

The following source data and figure supplements are available for figure 2:

**Source data 1.** Quantification of co-immunoprecipitation between p27 and Cortactin in MEF E6 (*Figure 2C*) and HeLa cells (*Figure 2—figure supplement 2*).

**Source data 2.** Statistical analyses for *Figure 2C* and *Figure 2—figure supplement 2*.

**Figure supplement 1.** p27 colocalizes with Cortactin and Tks5 in different tumor cell lines.

**Figure supplement 2.** p27/Cortactin interaction in HeLa cells after EGF stimulation.

To determine whether p27 affects invadopodia formation through the regulation of Cortactin, we tested if binding of p27 to Cortactin regulated the interaction of Cortactin with some of its partners. p27 overexpression in Hela cells had no effect on the interaction of Cortactin with c-Src, ERK1 or Arp2 (*Figure 5—figure supplement 1*) (*Kirkbride et al., 2011*). Also, Cortactin acetylation levels did not change when p27 was overexpressed (*Figure 5—figure supplement 1*), suggesting that p27 did not affect the interaction of Cortactin with acetyltransferases such as p300, Tip60 and ATAT1 or the deacetylases HDAC6 and Sirt1 (*Sun et al., 2015*; *Zhang et al., 2007*, *2009*; *Castro-Castro et al., 2012*). In contrast, the presence of p27 increased the amount of Cortactin co-immuno-precipitated with PAK1 in Hela cells (*Figure 5B*). Conversely, in MEFs lacking p27, the amount of Cortactin co-precipitated with PAK1 was sharply decreased compared to p27+/+ cells, despite the fact that PAK1 levels were elevated in p27−/− cells (*Figure 5C*). Monitoring the kinetics of association of PAK1 with Cortactin in MEFs revealed that some PAK1 is already associated with Cortactin in serum starved cells, consistent with a previous report (*Vidal et al., 2002*) and progressively increase after serum stimulation, reaching a maximal level after 3 hr (*Figure 5D and E*); mirroring the kinetics of association of p27 with Cortactin (*Figure 2C* and *Figure 2—figure supplement 2*). These results suggest that p27 binding to Cortactin promotes the association of PAK1 with Cortactin. Immunostaining of p27+/+ MEFs showed a colocalization of PAK1 and p27 at structures resembling invadopodia (*Figure 5F*). However, despite repeated attempts on both endogenous and overexpressed proteins, we were unable to obtain evidence supporting an interaction between p27 and PAK1 (data not shown).

## p27 regulates invadopodia formation via the Rac1/PAK1/phospho-Cortactin pathway

p27 promotes the interaction of PAK1 with Cortactin (*Figure 5A–D*), which in turn may promote invadopodia disassembly by phosphorylating Cortactin on Ser113 (*Moshfegh et al., 2014*). To test whether PAK1 activity is responsible for the low number of invadopodia observed in p27+/+ MEFs, we monitored the effect of either silencing PAK1 with siRNA or of the PAK1-3 specific inhibitors FRAX597 (*Licciulli et al., 2013*), FRAX1036 (*Ong et al., 2015*) and G-5555 (*Ndubaku et al., 2015*) on invadopodia formation and activity. While PAK1 silencing or the PAK1-3 inhibitors dramatically increased both the number of cells forming invadopodia and the area of matrix degraded per cell in p27+/+ MEFs, as reported previously (*Moshfegh et al., 2014*), it had no effect in p27−/− cells (*Figure 6A–B,D–F*, and *Figure 6—figure supplement 1A and B*). PAK1 silencing was verified by immunoblot for PAK1 in Control or PAK1 siRNA treated cells (*Figure 6C*). The efficacy of FRAX597, FRAX1036 and G-5555 in inhibiting PAK1 was evaluated by immunoblot on either vehicle or inhibitor treated cells using phospho-Ser144 PAK1/Ser141-PAK2 antibodies (*Figure 6G* and *Figure 6—figure supplement 1C*) (*Chong et al., 2001*). These results suggest that p27 acts downstream of PAK1 in

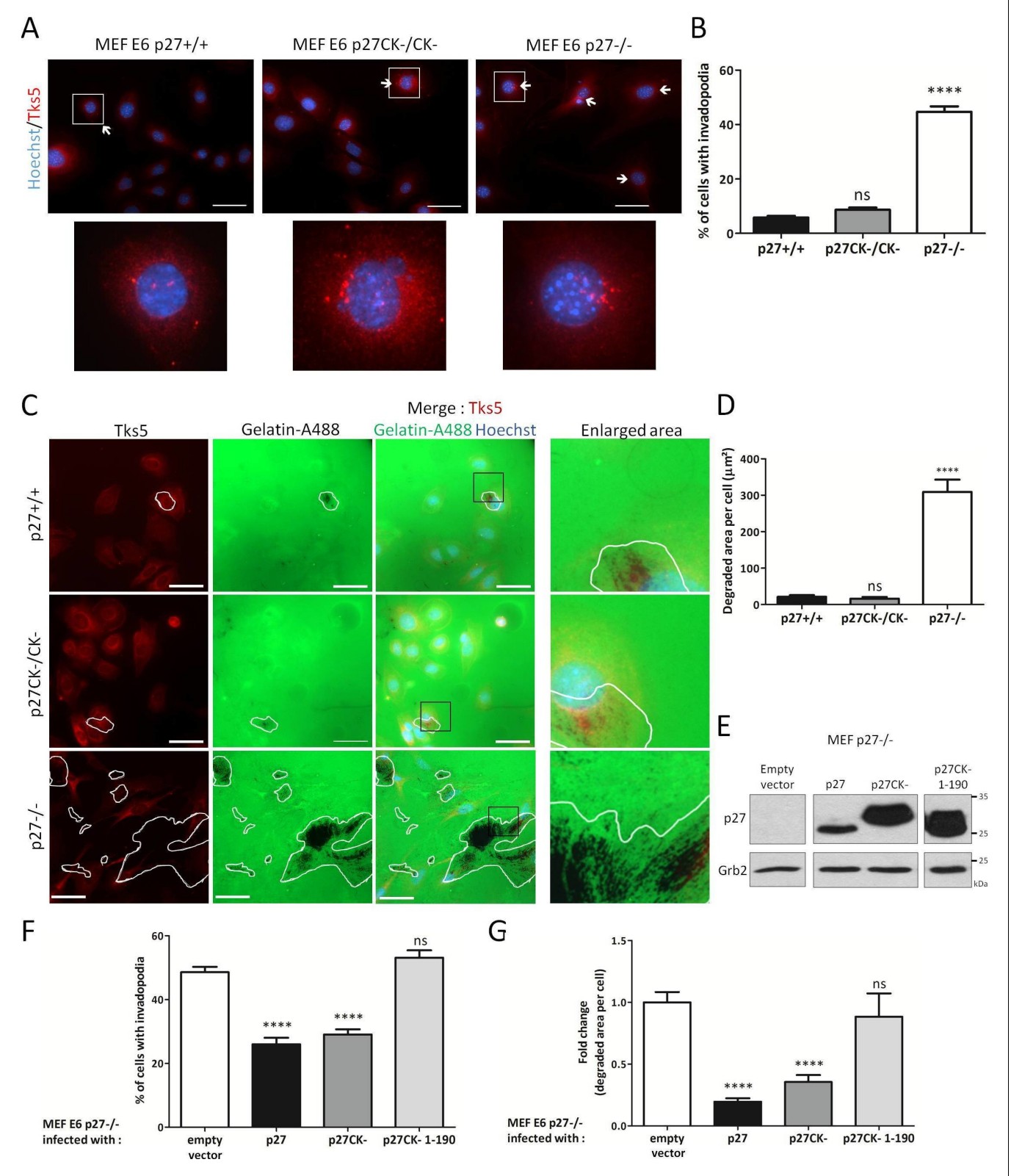

**Figure 3.** p27 regulates invadopodia formation and matrix degradation. (**A**) p27+/+, p27CK−/CK− and p27−/− immortalized MEFs were seeded on Oregon green-gelatin (gelatin-A488) for 16 hr. Cells were stained with rabbit anti-Tks5 (M-300) to visualize invadopodia. (**B**) The percentage of cells forming invadopodia was determined in a minimum of 15 fields, representing a minimum of 330 cells per genotype, for each experiment. The graph shows the mean of 3 independent experiments. (**C**) Cells were seeded as in (**A**). Tks5 staining shows invadopodia (red) and areas of degraded
*Figure 3 continued on next page*

*Figure 3 continued*

fluorescent gelatin indicate invadopodia activity (green). (D) The areas of degraded gelatin were measured in at least 15 fields per genotype in each experiment. The graph shows the mean of 3 independent experiments. (E–G) p27−/− E6 MEFs were infected with either empty vector or p27, p27CK− or p27CK− 1–190 vectors and then seeded on Gelatin-A488 for 48 hr. (E) p27 levels after retroviral infection were determined by immunoblot with rabbit anti-p27 (N-20) antibodies; Grb2 was used as loading control. (F–G) After Tks5 staining, invadopodia forming cells (F) were quantified as in B and area of degraded gelatin (G) as in D. Scale bars: 50 μm; 'ns' not significant; ****p<0.0001. In A and C, images were acquired using a 40x objective and images displayed are cropped areas.

The following source data and figure supplements are available for figure 3:

**Source data 1.** Quantification of cells with invadopodia (*Figure 3B*); quantification of degraded gelatin area per cell (*Figure 3C*); quantification of cells with invadopodia after p27 re-expression (*Figure 3F*) and quantification of degraded gelatin area per cell after p27 re-expression (*Figure 3G*).

**Source data 2.** Statistical analyses for *Figure 3B,C,F and G*.

**Source data 3.** Immunoblot scans of *Figure 3E*.

**Figure supplement 1.** p27+/+, p27CK−/CK− and p27−/− MEFs form functional invadopodia.

**Figure supplement 2.** p27 colocalizes with Tks5 at sites of gelatin degradation.

the regulation of invadopodia formation and activity, and that this pathway is impaired in p27−/− cells, possibly due to decreased PAK1/Cortactin interaction.

If p27 acts on the invadopodia disassembly pathway at the level of PAK1/Cortactin, one would not expect to see any difference in Rac1 activation levels in p27+/+ and p27−/− cells. This is what was observed by GTP-Rac1 pull-down assays (*Figure 7A*), in agreement with our previous observations (*Besson et al., 2004b*). Nevertheless, to further characterize how the Rac1/PAK1/Cortactin signaling pathway is affected in function of p27 status, we inhibited Rac1 itself using either Rac1 siRNAs (*Figure 7B–D*) or the Rac specific inhibitor NSC23766 (*Gao et al., 2004*) (*Figure 7E and F*). Rac1 silencing by siRNAs was verified by immunoblot for Rac1 (*Figure 7D*). When Rac1 was silenced or inhibited, more p27+/+ cells formed invadopodia and there was a dramatic increase in the area of matrix degraded (*Figure 7B,E and C,F*, respectively), as reported previously (*Moshfegh et al., 2014*). On the other hand, in p27−/− cells, Rac1 silencing or inhibition had no effect on invadopodia formation or gelatin degradation (*Figure 7B,E and C,F*, respectively); supporting the idea that p27 regulates invadopodia disassembly downstream of Rac1 and PAK1.

We previously reported that p27 binds RhoA and prevents its activation by its GEFs; consequently, cells lacking p27 have elevated RhoA activity (*Besson et al., 2004b*). Since RhoA plays a critical role in cell motility and the control of actin cytoskeleton dynamics, we determined whether RhoA was involved in p27-mediated regulation of invadopodia formation and function. Transfection of RhoA siRNAs in p27+/+ and p27−/− MEFs did not alter the proportion of cells forming invadopodia (*Figure 7—figure supplement 1A*) and only mildly increased the ability of both p27+/+ and p27−/− cells to degrade ECM, although this was not statistically significant (*Figure 7—figure supplement 1B*). RhoA siRNA efficacy was checked by immunoblot (*Figure 7—figure supplement 1C*). Furthermore, inhibition of the Rho effectors ROCK1/2 with a pharmacological inhibitor (Y27632) did not affect the ability of p27+/+ and p27−/− cells to invade through Collagen I in transwell invasion assays (*Figure 7—figure supplement 1D*). Thus, in MEFs, depletion of RhoA did not affect the ability of cells to form invadopodia and slightly increased invadopodia proteolytic activity independently of p27 status.

Evidence suggests that invadopodia disassembly by the Trio/Rac1/PAK1 signaling pathway is mediated by phosphorylation of Cortactin on Ser113 (*Webb et al., 2005*; *Moshfegh et al., 2014*). In the latter study, expression of an unphosphorylatable mutant of Cortactin (Cortactin S113A) increased both invadopodia lifetime and matrix degradation, as observed upon Rac1 or PAK1 silencing (*Moshfegh et al., 2014*). We retrovirally infected p27+/+ and p27−/− E6 MEFs with either empty vector, WT Cortactin, Cortactin S113A or the phosphomimetic mutant Cortactin S113D (*Figure 8A*) and monitored their effect on invadopodia formation and activity. In p27+/+ cells,

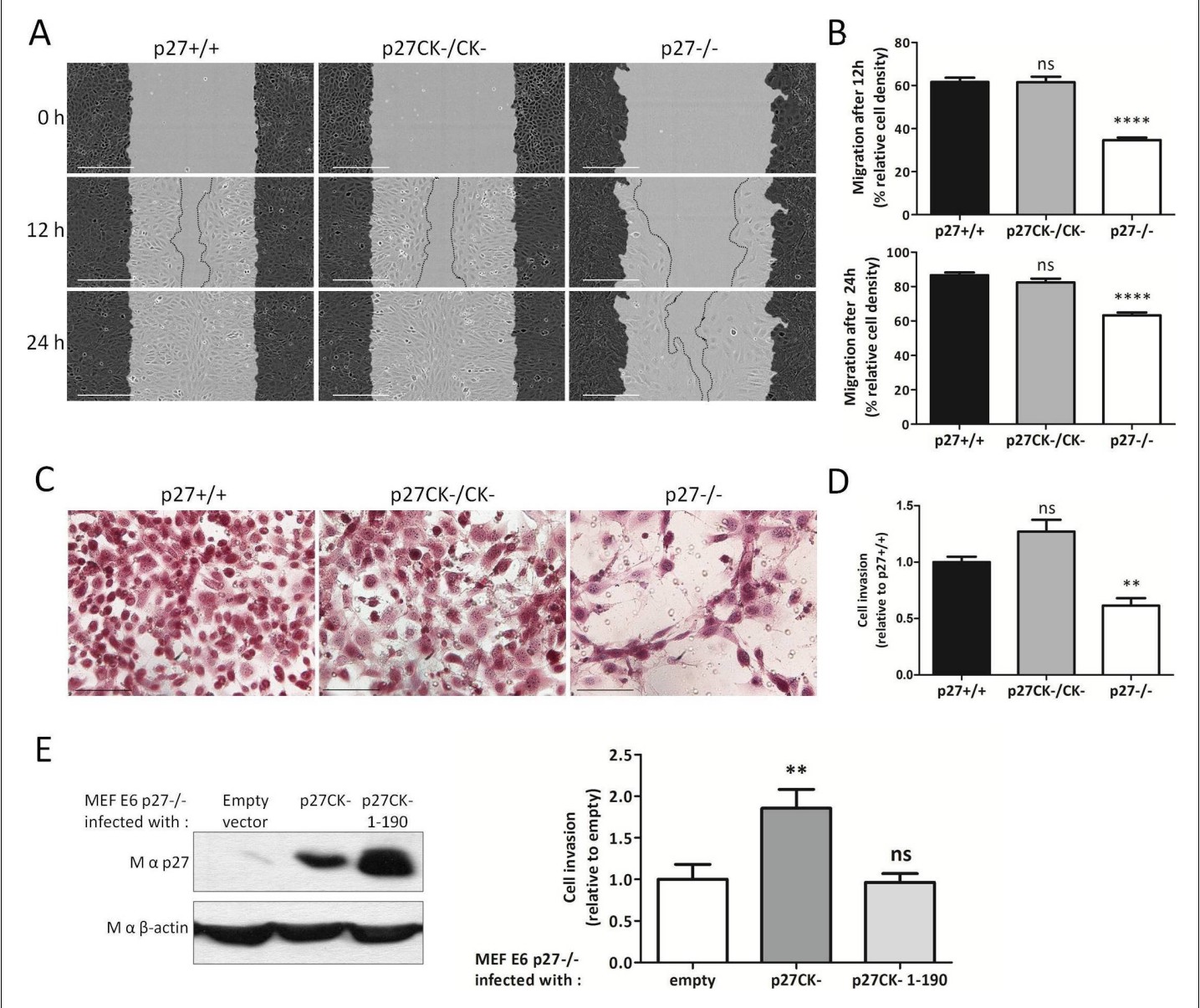

**Figure 4.** p27 promotes cell migration and invasion. (**A**) Representative images of scratch wound migration assays with p27+/+, p27CK−/CK− and p27−/− immortalized MEFs. Dark grey areas show the initial wound masks and dotted lines the migration fronts. Scale bars: 300 µm. (**B**) Mean cell migration at 12 hr and 24 hr post wounding for each genotype of five independent experiments. Percent of area in which the cells migrated, or wound closing (relative wound density) was calculated with the Incucyte software. 'ns': not significant; ****p<0.0001. (**C**) Representative images of p27+/+, p27CK−/CK− and p27−/− immortalized MEFs that invaded through a layer of Collagen I in transwell invasion assays and migrated to the bottom side of the transwell membrane after 48 hr. Scale bars: 100 µm. (**D**) The graph shows the mean number of cells that invaded through Collagen I quantified by XTT staining, expressed relative to p27+/+ cells, of three independent experiments. **p<0.01. (**E**) p27−/− E6 MEFs were infected with either empty vector, p27CK- or p27CK- 1–190 vectors and used in transwell invasion assays as in (**C–D**). p27 levels after retroviral infection were determined by immunoblot with mouse anti-p27 (SX53G8.5); β-actin was used as loading control. The graph shows the mean number of cells that invaded through Collagen I quantified by XTT staining, expressed relative to p27−/− cells, of four independent experiments.

The following source data is available for figure 4:

**Source data 1.** quantification of relative wound density (*Figure 4B*); quantification of invasion (*Figure 4D*); and quantification of invasion rescue by p27 re-expression (*Figure 4E*).
**Source data 2.** Statistical analyses for *Figure 4B,D and E*.

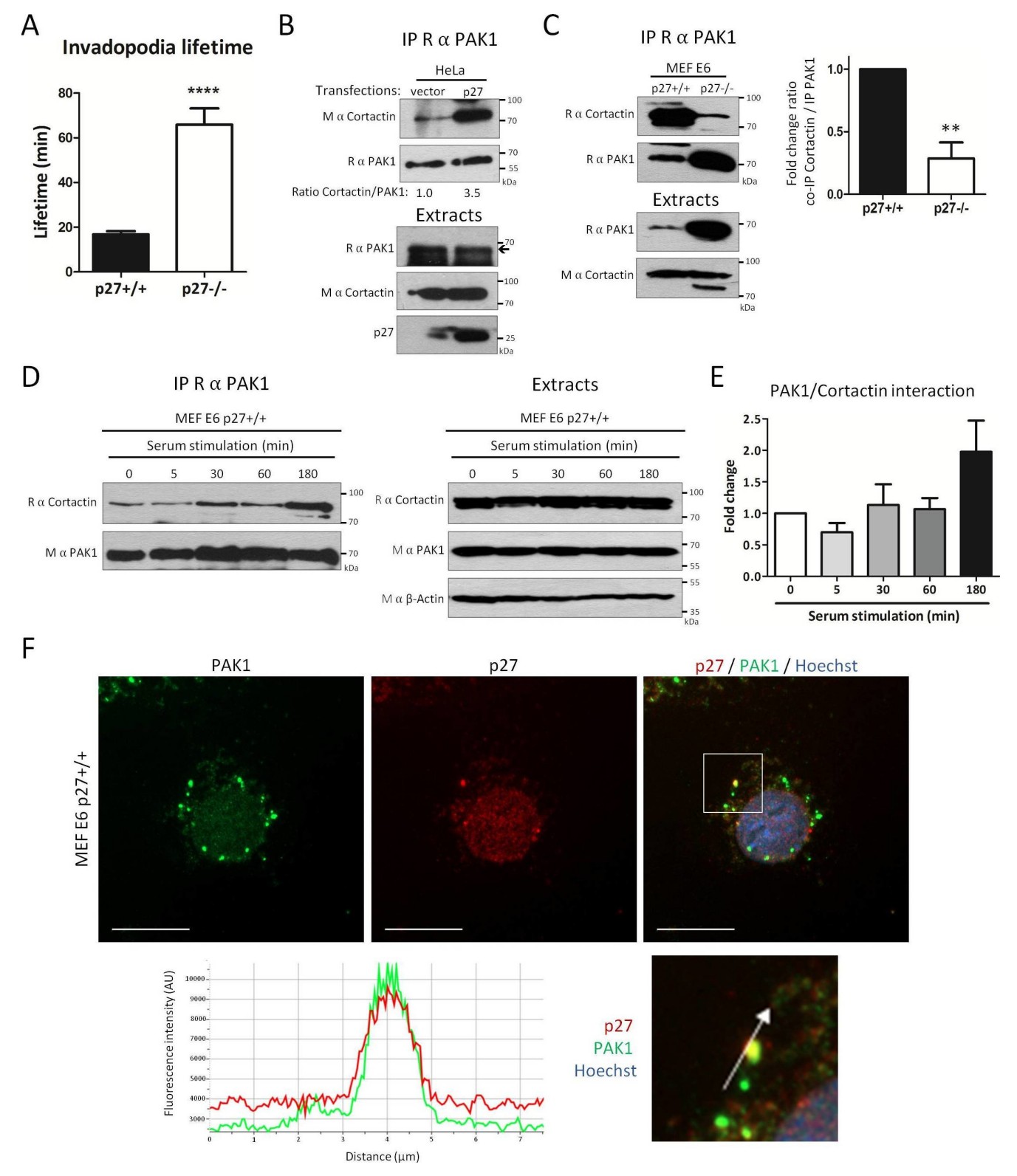

**Figure 5.** p27 promotes binding of Cortactin to PAK1. (**A**) Live p27+/+ and p27−/− immortalized MEFs expressing eGFP-Tks5 were imaged by videomicroscopy to measure invadopodia lifetime, using Tks5 as an invadopodia marker. The graph represents the average invadopodia lifetime of 20 invadopodia per genotype per experiment from three independent experiments. (**B–C**) Co-immunoprecipitations using rabbit anti-PAK1 (N-20) of HeLa cell lysates transfected with empty vector or p27 encoding vector (**B**) or p27+/+ and p27−/− E6 MEF lysates (**C**). Co-immunoprecipitated Cortactin was

*Figure 5 continued on next page*

*Figure 5 continued*

detected with mouse (**B**) or rabbit (**C**) anti-Cortactin antibodies. Immunoprecipitated PAK1 was visualized by reprobing the membranes with rabbit anti-PAK1 (N-20) and anti-Rabbit Ig light-chain secondary antibodies. Immunoblots of extracts show the level of proteins in each condition. In (**C**), the graph shows the mean fold change in the ratio of Cortactin to PAK1 co-immunoprecipitated, expressed relative to p27+/+ cells, in three independent experiments. **p<0.01. (**D**) p27+/+E6 MEFs were starved overnight in DMEM 0.1% FCS and then stimulated with growth medium for the indicated times. PAK1 was immunoprecipitated from cell lysates using rabbit anti-PAK1 (N-20). Immunoblots of immunoprecipitates (left panels) and extracts (right panels) were probed successively with rabbit anti-Cortactin (H-191) and anti-rabbit Ig light chain secondary antibodies and then with mouse anti-PAK1 (A6) antibodies. β-actin was used as loading control. (**E**) The graph shows the mean amount of Cortactin bound to PAK1 at each time-point from four independent experiments, normalized to time zero. These differences were not statistically significant. (**F**) p27+/+ MEFs were seeded on coverslips and permeabilized with digitonin prior to fixation. Cells were labeled with rabbit anti-PAK1 (N-20, green) and mouse anti p27 (SX53G8.5, red) antibodies. Images were acquired using a 60x objective and images displayed are cropped areas. The graph displaying the fluorescence intensity (arbitrary unit) under the arrow in the enlarged panel was generated with NIS Element software. Scale bars: 20 μm.

The following source data and figure supplement are available for figure 5:

**Source data 1.** Quantification of invadopodia lifetime (*Figure 5A*); quantification of co-immunoprecipitation between Cortactin and PAK1 in MEFs (*Figure 5C*); and quantification of co-immunoprecipitation between Cortactin and PAK1 after serum stimulation (*Figure 5E*).
**Source data 2.** Statistical analyses for *Figure 5A*.
**Source data 3.** Statistical analyses for *Figure 5C*.
**Source data 4.** Statistical analyses for *Figure 5E*.
**Figure supplement 1.** p27 does not affect Cortactin acetylation or the recruitment of c-Src, Arp2 and ERK1 to Cortactin.

Cortactin S113A increased the number of cells forming invadopodia, as observed previously (*Moshfegh et al., 2014*), while Cortactin S113D had no effect (*Figure 8B and D*). In contrast, Cortactin S113A had no effect in p27−/− cells, confirming that the Rac1/PAK1/Cortactin pathway is defective in absence of p27 and that p27 acts upstream of Cortactin. On the other hand, expression of Cortactin S113D in p27−/− cells decreased the number of invadopodia forming cells and the area of degraded gelatin (*Figure 8C and E*), indicating that mimicking Cortactin phosphorylation rescues the phenotype caused by the absence of p27.

Importantly, PAK3 was reported to phosphorylate Cortactin on several serine residues (S113, S150 and S282) within the actin binding repeats (*Webb et al., 2006*) suggesting that multiple phosphorylation events may synergize to regulate Cortactin function. To test this possibility, we generated triple alanine mutant (TA) (S113A/S150A/S282A) and triple aspartic acid mutant (TD) (S113D/S150D/S282D) of Cortactin and expressed these constructs in p27+/+ and p27−/− E6 MEFs (*Figure 8F*). Cortactin TA and TD mutants gave essentially similar results as the Cortactin S113A and S113D mutants, respectively (*Figure 8G–J*). Nevertheless, in p27+/+ MEFs, the Cortactin TA mutant was more potent than Cortactin S113A to promote invadopodia formation (13.5% p27+/+ cells forming invadopodia with Cortactin S113A versus 23.3% with Cortactin TA) and activity (6.17 fold increase in area of degraded gelatin with Cortactin S113A versus 10.02 fold with Cortactin TA) (*Figure 8B* versus *Figure 8G* and *Figure 8D* versus *Figure 8I*, respectively). This data suggests that Cortactin phosphorylation on S150 and/or S282 also contribute with S113 to control invadopodia disassembly. To further confirm that PAK-mediated phosphorylation events on Cortactin are involved in this pathway, we evaluated Cortactin phosphorylation levels in HEK 293 cells expressing Myc-tagged Cortactin treated or not with the PAK1-3 inhibitor FRAX597. In presence of FRAX597, there was a 36% decrease in P-Ser Cortactin level (*Figure 8—figure supplement 1A and B*). This mild reduction in P-Ser levels is consistent with Cortactin also being a substrate for other Ser/Thr kinases such as ERK, Akt and PKC. The contribution of S113, S150 and S282 phosphorylation in the P-Ser signal observed in Cortactin immunoprecipitates was estimated in HEK 293 cells expressing either wild-type Cortactin, Cortactin S113A or Cortactin TA. P-Ser levels were reduced in Cortactin S113A immunoprecipitations compared to WT Cortactin and further decreased in Cortactin TA immunoprecipitates, confirming that phosphorylation of these sites contribute to the Cortactin P-Ser signal in vivo (*Figure 8—figure supplement 1C*). Finally, to show that Cortactin was phosphorylated

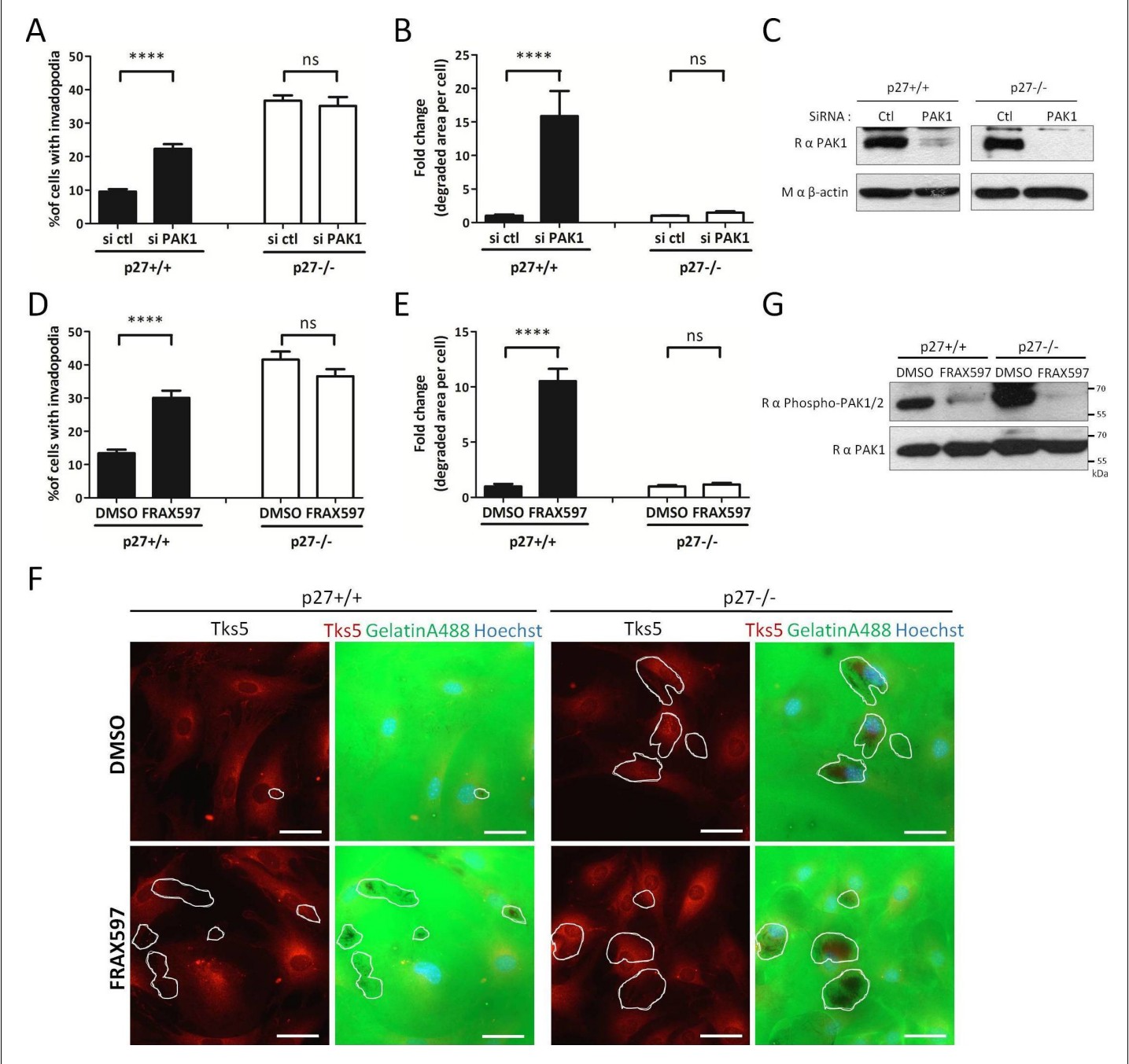

**Figure 6.** p27 regulates invadopodia formation downstream of PAK1. (**A–B**) p27+/+ and p27−/− E6 MEFs were seeded on gelatin-A488 and transfected with either control siRNAs or PAK1 siRNAs for 48 hr. Cells were stained with rabbit anti-Tks5 (M-300) to visualize invadopodia. At least ten fields per conditions were used to count cells forming invadopodia (**A**), representing a minimum of 212 cells per genotype for each experiment, or to measure the area of degraded gelatin, expressed in fold-change compared to vehicle treated conditions (**B**). (**C**) PAK1 siRNA efficacy at 48 hr was evaluated by immunoblot with rabbit anti-PAK1 (N-20) antibodies. $\beta$-actin was used as loading control. (**D–F**) p27+/+ and p27−/− E6 MEFs were seeded for 48 hr on gelatin-A488 and after 1 hr were treated with DMSO or 1 µM FRAX597, a PAK1-3 inhibitor. Quantification of cells forming invadopodia (**D**) and gelatin degradation (**E**) was performed as in (**A–B**), with a minimum of 215 cells counted per genotype per experiment. (**F**) Images were acquired using a 40x objective and images displayed are cropped areas. Scale bars: 50 µm. (**G**) FRAX597 inhibitor efficacy was evaluated by immunoblot with rabbit anti-phospho-Ser144-PAK1/phospho-Ser141-PAK2 and rabbit anti-PAK1 (N-20) antibodies. (**A–B**; **D–E**) The graphs show the mean of at least three independent experiments. ****p<0.0001.

The following source data and figure supplement are available for figure 6:

*Figure 6 continued on next page*

Figure 6 continued

**Source data 1.** Quantification of invadopodia forming cells (*Figure 6A*) and degraded gelatin area (*Figure 6B*) after PAK1 silencing; quantification of invadopodia forming cells (*Figure 6D*) and degraded gelatin area (*Figure 6E*) after FRAX597 treatment; quantification of invadopodia forming cells (*Figure 6—figure supplement 1A*) and degraded gelatin area (*Figure 6—figure supplement 1B*) after FRAX1036 and G-5555 treatment.
**Source data 2.** statistical analyses for *Figure 6A,B,D and E* and *Figure 6—figure supplement 1A and B*.
**Figure supplement 1.** p27 regulates invadopodia formation in a PAK1 dependent manner.

in vivo on PAK-targeted sites (S113, S150 and/or S282), we performed mass spectrometry analyses on Cortactin. For this, HEK 293 cells were transfected with Myc-Cortactin alone or Myc-Cortactin and the myristoylated second SH3 domain of Nck1 (Myr-SH3-2), which was previously shown to activate PAK1 (*Lu et al., 1997*; *Lu and Mayer, 1999*). MS/MS analyses from Cortactin immunoprecipitates indicated that Cortactin was phosphorylated on S150 (*Figure 8—figure supplement 2*). This phosphorylation was detected only in samples from cells co-expressing Cortactin and Myr-SH3-2 suggesting that PAK1 activation is required for this event. Although no phosphorylation was detected on S113 or S282 in these experiments, phosphoproteome analyses from other studies have previously shown that Cortactin is phosphorylated in vivo on S113, S150 and/or S282: S113 phosphorylation was detected in prostate cancer cells (*Chen et al., 2010*), mouse brain (*Wiśniewski et al., 2010*) and breast cancers (*Mertins et al., 2016*). Phosphorylation on S150 was identified in mouse renal cells and human liver (*Bian et al., 2014*; *Rinschen et al., 2010*) and in the present study. S282 phosphorylation was detected in mitotic cells and breast cancers (*Mertins et al., 2016*; *Olsen et al., 2010*; *Klammer et al., 2012*). Phosphorylation on all three sites have been detected in ovarian and breast cancer xenogratfs (*Mertins et al., 2014*).

Altogether, our results indicate that p27 promotes invadopodia turnover and cell invasion via the Rac1/PAK1/Phospho-Cortactin pathway. In contrast, in absence of p27, this pathway is largely defective, resulting in increased invadopodia stability and elevated matrix degradation, but impaired invasion.

## Discussion

Our results indicate that p27 regulates invadopodia formation and activity by binding to Cortactin, probably via the regulation of invadopodia turnover mediated by a signaling pathway recently characterized whereby signaling through Trio/Rac1/PAK1/Cortactin induces the disassembly of invadopodia (*Moshfegh et al., 2014*). The main finding presented here is that p27 directly controls cellular invasion by impinging on this pathway through the promotion of the Cortactin/PAK1 interaction and the phosphorylation of Cortactin on Ser113, S150 and/or S282 by PAK. Indeed, inhibiting the pathway upstream of Cortactin, either with Rac1 and PAK1 inhibitors or siRNAs converted the phenotype of p27+/+ cells to that of p27−/− cells. Conversely, artificially activating this pathway at the level of Cortactin using phosphomimetic mutants for the PAK-targeted sites efficiently rescued the phenotype of p27-null cells. All three phosphorylation sites on Cortactin appear to act synergistically as the phenotype caused by the triple mutants (Cortactin TA and TD) was more profound than that of mutating S113 alone. While mimicking the phosphorylation of Cortactin on PAK1-targeted sites clearly causes the disassembly of invadopodia (*Moshfegh et al., 2014*; *Webb et al., 2005*), the mechanism involved is still unclear but may be due to decreased affinity of Cortactin phosphorylated within its Actin-binding repeats for F-Actin, which may destabilize invadopodia (*Webb et al., 2006*).

Results from *Moshfegh et al. (2014)* and the present study show that inhibition of the Trio/Rac1/PAK1/Phospho-Cortactin pathway stabilizes invadopodia and is associated with a dramatic increase in proteolytic degradation of ECM. Surprisingly, this phenotype is accompanied by a sharp decrease in the ability of cells to invade through matrix in 3D invasion assays. Thus, it appears that the turnover of invadopodia rather than elevated ECM degradation is required for efficient invasion. Affecting the dynamics of invadopodia formation, maturation and disassembly is therefore detrimental to invasion, as observed here when this last step is inhibited by interfering with Rac1/PAK1 signaling or when p27 is absent. Interestingly, p27 also regulates cell migration in a similar manner by interfering

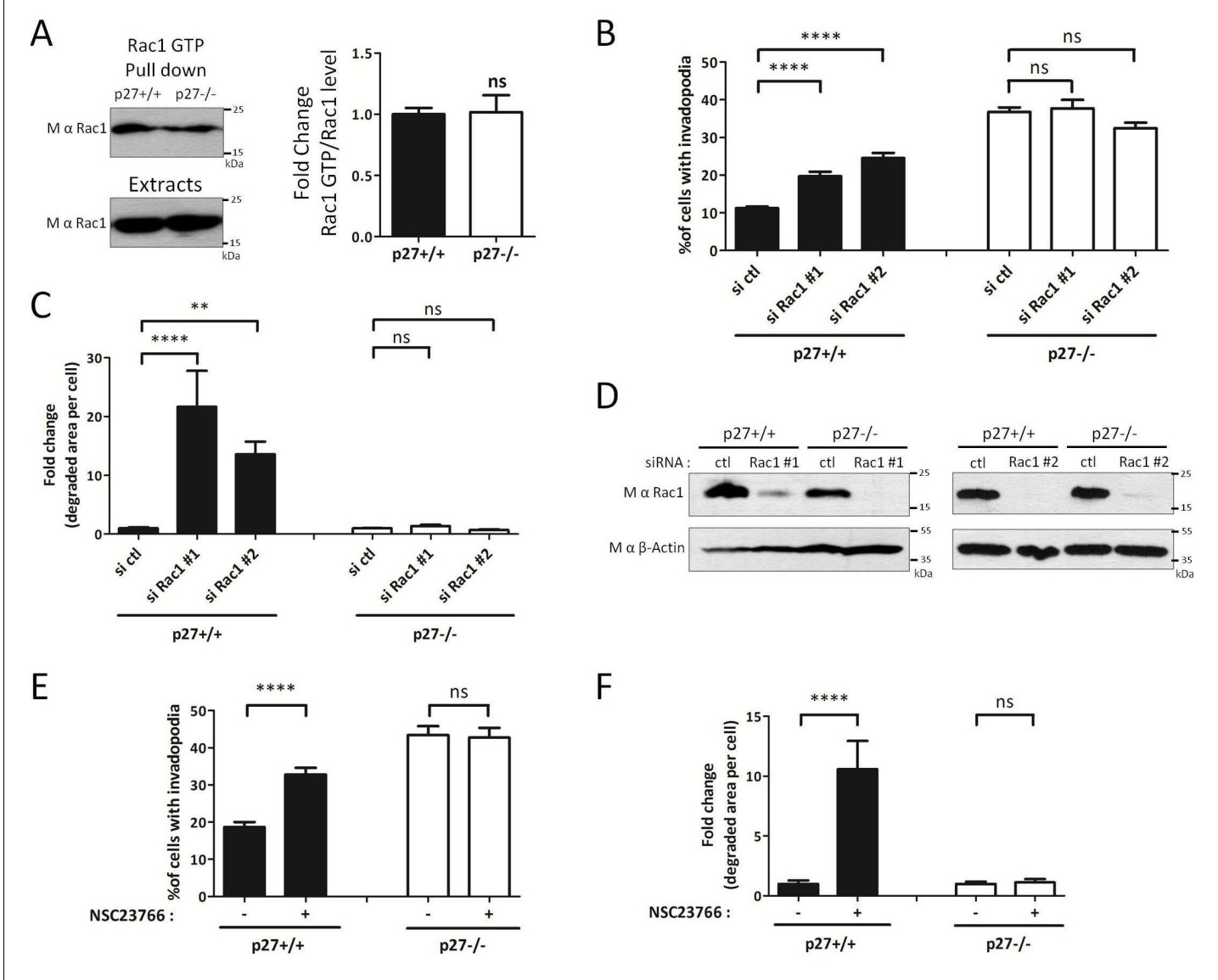

**Figure 7.** p27 regulates invadopodia formation downstream of Rac1. (**A**) Cells were seeded for 24 hr and Rac1-GTP levels measured by GTP pull-downs assays using GST-PAK1-CD beads. The amounts of Rac1-GTP bound to the beads and of total Rac1 in the extracts were detected by immunoblot with mouse anti-Rac1. The graph shows the mean ratio of GTP-Rac1/total Rac1 from six independent experiments. (**B–D**) Cells were transfected with control (ctl) or Rac1 #1 or #2 siRNAs for 3 days. Cells were then seeded on Gelatin-A488 for 48 hr and for monitoring siRNA efficiency. Cells were stained with rabbit anti-Tks5 (M-300) to visualize invadopodia. At least ten fields per condition were used to count cells forming invadopodia, representing a minimum of 197 cells per genotype for each experiment (**B**) or to measure the area of degraded gelatin, expressed in fold-change compared to control siRNA treated conditions (**C**). (**D**) Rac1 silencing was evaluated by immunoblot with mouse anti-Rac1. $\beta$-actin was used as loading control. (**E–F**) Cells were processed as described in **Figure 6D and E** except that the Rac1 inhibitor NSC23766 at 100 µM was used instead of FRAX597. A minimum of 153 cells were counted per genotype for each experiment. (**B–C**; **E–F**) The graphs show the mean of at least three independent experiments. ****p<0.0001; **p<0.01.

The following source data and figure supplement are available for figure 7:

**Source data 1.** Quantification of Rac1 GTP/Rac1 levels (**Figure 7A**); quantification of invadopodia forming cells (**Figure 7B**) and degraded gelatin area (**Figure 7C**) after silencing of Rac1; quantification of invadopodia forming cells (**Figure 7E**) and degraded gelatin area (**Figure 7F**) after NSC23766 treatment; quantification of invadopodia forming cells (**Figure 7—figure supplement 1A**) and degraded gelatin area (**Figure 7—figure supplement 1B**) after RhoA silencing; and quantification of invasion after Y27632 treatment (**Figure 7—figure supplement 1D**).

**Source data 2.** Statistical analyses for **Figure 7A,B,C,E,F**, and **Figure 7—figure supplement 1A,B and D**.

*Figure 7 continued on next page*

*Figure 7 continued*

**Figure supplement 1.** RhoA regulation by p27 is not involved in invadopodia formation.

with RhoA activation by its GEFs (*Besson et al., 2004a*, *2004b*). In absence of p27, cells accumulate actins stress fibers and focal adhesions and exhibit a migration defect due to an imbalance in the dynamic cycles of RhoA activation and inactivation (and concomitant Rac1 inactivation and activation) required for F-actin turnover, focal adhesion disassembly and efficient migration (*Besson et al., 2004a*, *2004b*; *Ren et al., 2000*; *Arthur and Burridge, 2001*; *Cox et al., 2001*; *Sahai et al., 2001*; *Vial et al., 2003*). Migration and invasion are two intimately linked processes (*Friedl and Wolf, 2003*; *Petrie and Yamada, 2016*). It is quite remarkable that p27 appears to regulate both of these processes, although by distinct mechanisms. Indeed, p27 binds to RhoA and Cortactin via the same domain located within the last 8 aa of the protein (*Figure 1E*) (*Larrea et al., 2009*; *Godin et al., 2012*), but while RhoA preferentially binds p27 phosphorylated on T198 (*Larrea et al., 2009*), the interaction of p27 with Cortactin did not require the presence of T198 (*Figure 1E*), suggesting that different pools of the protein are involved in their regulation. In addition, the regulation of invadopodia by p27 did not appear to involve RhoA, as RhoA silencing affected both p27−/− and p27+/+ cells in a similar manner (*Figure 7—figure supplement 1*). p27 also regulates invasion by promoting EMT, notably through JAK2/STAT3 signaling and induction of the transcription factor Twist (*Zhao et al., 2015*).

We found that p27 binds to Cortactin on its actin-binding repeats and this interaction promoted the association of Cortactin with PAK1. Since we were unable to detect an interaction between p27 and PAK1 using either endogenous or overexpressed proteins, it remains unknown how p27 promotes the PAK1/Cortactin interaction. Evidence suggest that Cortactin is an intrinsically disordered protein (IDP), especially its actin-binding repeats (*Shvetsov et al., 2009*) and p27 is a well-known IDP (*Galea et al., 2008*; *Lacy et al., 2004*; *Ou et al., 2012*). Since these proteins typically fold upon binding to their partners, an attractive possibility is that p27 binding causes a conformation change in Cortactin that would favor its interaction with PAK1.

p27 is a cyclin-CDK inhibitor and although p27 regulates migration and invasion in a CDK-independent manner (*Besson et al., 2004b*; *Larrea et al., 2009*; *Zhao et al., 2015*; *Gui et al., 2014*), there is evidence to suggest that some cell cycle independent roles of CDKs and cyclins may be mediated via CDK inhibitors such as p27 (*Hydbring et al., 2016*). For instance, the regulation of cell migration by Cyclin D1 requires p27 (*Li et al., 2006*) and the role of CDK5 in promoting neuronal progenitor migration in vivo is mediated via the phosphorylation of p27 on Ser10 which regulates the RhoA/ROCK/LIMK/Cofilin pathway (*Nguyen et al., 2006*; *Kawauchi et al., 2006*). Interestingly, CDK5 was recently found to play an important role in invadopodia formation and cancer cell invasion, and CDK5 inhibition blocked both these processes (*Quintavalle et al., 2011*; *Bisht et al., 2015*). CDK5 phosphorylated Caldesmon, causing its degradation by the proteasome and promoting invadopodia formation and invasion (*Quintavalle et al., 2011*). Since p27 is a poor inhibitor of CDK5 complexes (*Lacy et al., 2005*; *Lee et al., 1996*), it would be interesting to test whether this role of CDK5 may involve p27. Indeed, an attractive hypothesis is that p27 could simultaneously interact with CDK5 complexes, via its N-terminal cyclin-CDK binding domain, and with Cortactin via its C-terminus, thereby recruiting CDK5 in proximity to Caldesmon bound to Cortactin and allowing Caldesmon phosphorylation.

Our study further defines the complex role played by p27 in the regulation of cytoskeletal dynamics, migration, EMT and invasion and provides another element to explain why the *Cdkn1b* gene is rarely mutated in cancer (*Chu et al., 2008*; *Besson et al., 2008*; *Kandoth et al., 2013*). Indeed, p27 is either downregulated, mostly via increased proteasomal degradation, or excluded from the nuclei of cancer cells. Given that upon cytoplasmic relocalization, p27 promotes both migration and invasion and may serve to coordinately regulate these processes, it appears likely that this feature may be selected for during tumor progression and could participate in the acquisition of a metastatic behavior by cancer cells.

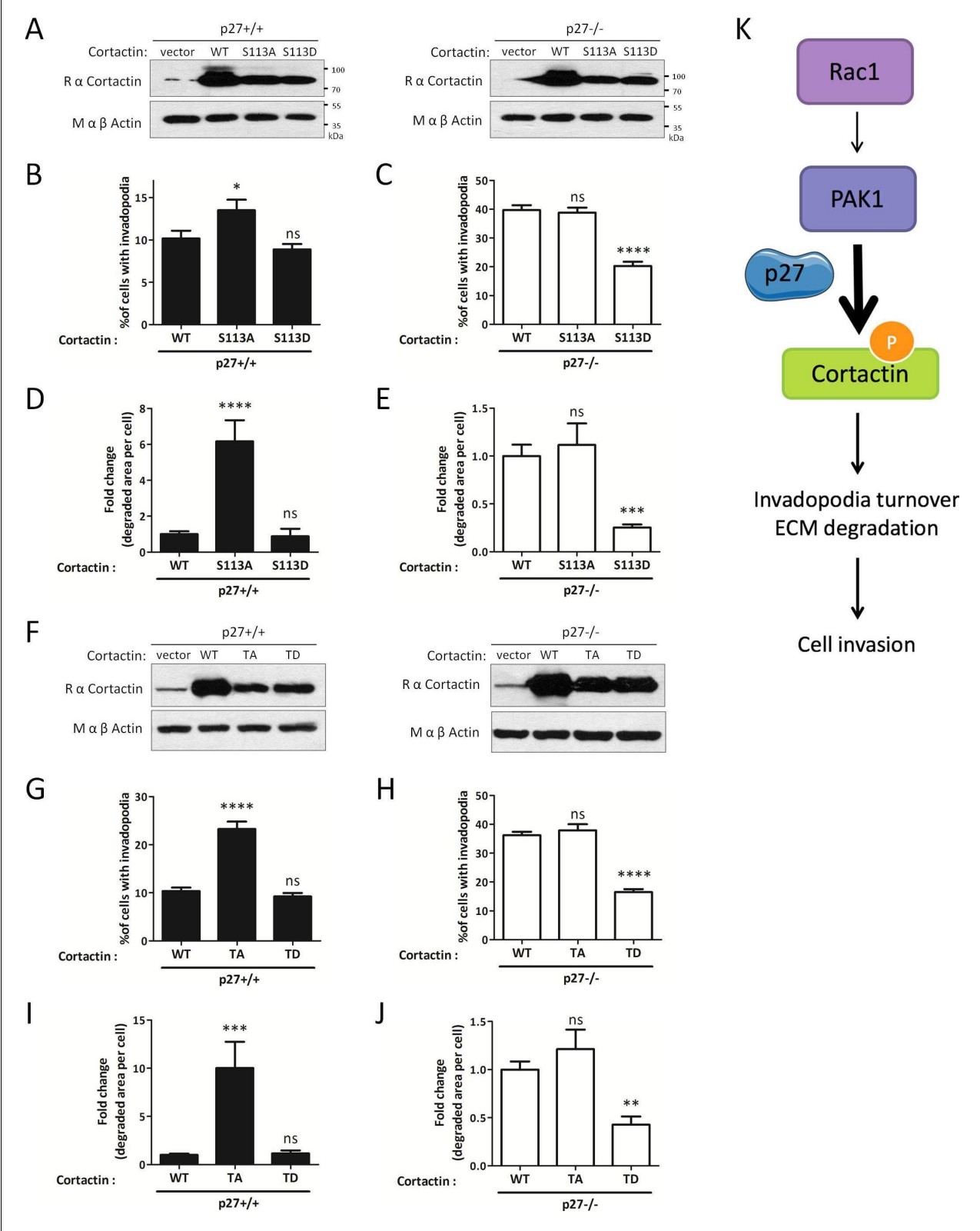

**Figure 8.** Mimicking phosphorylation of Cortactin restores invadopodia dynamics in p27−/− cells. (A–E) p27−/− E6 MEFs were infected with empty vector or with vectors encoding wild type Cortactin (WT), S113A-Cortactin (S113A) or S113D-Cortactin (S113D). (A) Cortactin levels after retroviral infection were determined by immunoblot with rabbit anti-Cortactin (H-191) antibodies. β-actin was used as loading control. (B–E) Cells were seeded on gelatin-A488 for 48 hr. After Tks5 staining, cells forming invadopodia (B–C), or the area of degraded gelatin, expressed in fold-change compared to

*Figure 8 continued on next page*

*Figure 8 continued*

WT Cortactin transfected cells (**D–E**), were quantified in at least ten fields per condition in each experiment, representing a minimum of 179 cells per genotype. The graphs show the means of at least three independent experiments. (**F–J**) p27−/− E6 MEFs were infected with empty vector or with vectors encoding WT Cortactin, Cortactin TA (S113A/S150A/S282A) or Cortactin TD (S113D/S150D/S282D). (**J**) Cortactin levels after retroviral infection were determined as in (**A**). (**G–J**) Cells were processed as in (**B–E**) to quantify cells forming invadopodia (**G–H**), or the area of degraded gelatin (**I–J**), with a minimum of 222 cells counted per genotype per experiment. The graphs show the means of 3 independent experiments. (**K**) Schematic representation of the Rac1/PAK1/phospho-Cortactin pathway involved in invadopodia turnover and matrix degradation and its proposed regulation by p27. ****$p<0.0001$; ***$p<0.001$; **$p<0.01$; *$p<0.05$.

The following source data and figure supplements are available for figure 8:

**Source data 1.** Quantification of cells forming invadopodia (***Figure 8B–C***) and degraded gelatin area (***Figure 8D–E***) after infection with S113 phospho-mutants of Cortactin; quantification of cells forming invadopodia (***Figure 8G–H***) and degraded gelatin area (***Figure 8I–J***) after infection with triple phospho-mutants of Cortactin; quantification of P-Ser Cortactin/Cortactin ratio (***Figure 8—figure supplement 1B***).
**Source data 2.** Statistical analyses for ***Figure 8***.
**Source data 3.** Statistical analyses for ***Figure 8—figure supplement 1B***.
**Source data 4.** Mascot search results for Cortactin for ***Figure 8—figure supplement 2***.
**Figure supplement 1.** Cortactin is phosphorylated on S113/S150 and/or S282 in vivo.
**Figure supplement 2.** Cortactin is phosphorylated on S150 in vivo upon PAK activation.

# Materials and methods

## Antibodies, reagents and plasmids

Mouse anti c-Myc (9E10, sc-40, RRID:AB_627268), p27 (F8, sc-1641, RRID:AB_628074), p27 (SX53G8.5, sc-53871, RRID:AB_785029), αPAK (A6, sc-166887, RRID:AB_10609226), RhoA (26C4, sc-418, RRID:AB_628218) and rabbit anti p27 (C19, sc-528, RRID:AB_632129), Myc (A14, sc-789, RRID: AB_631274), Cortactin (H191, sc-11408, RRID:AB_2088281), Tks5 (M-300, sc-30122, RRID:AB_ 2254551), αPAK (N-20, sc-882, RRID:AB_672249), Arp2 (H-84, sc-15389, RRID:AB_2221848), c-Src (SRC2, sc-18, RRID:AB_631324) and ERK1 (K-23, sc-94, RRID:AB_2140110) antibodies were purchased from Santa Cruz Biotechnology. Mouse anti p27 (610242), Grb2 (610112, RRID:AB_397518), Cortactin (610050, RRID:AB_397462), phopsho-Ser (612547, RRID:AB_399842) and Rac1 (610650, RRID:AB_397977) antibodies were purchased from BD-Transduction Laboratories. Mouse anti β-actin (A2228, RRID:AB_476697) antibody was purchased from Sigma-Aldrich. Rabbit anti phospho-Ser144-PAK1/phospho-Ser141-PAK2 (#2606, RRID:AB_2299279) antibody was purchased from Cell Signalling Technology. Rabbit anti acetyl-Cortactin (#09–881, RRID:AB_10584980) antibody was purchased from Millipore. Secondary antibodies against whole Ig or Ig light-chain conjugated to horseradish peroxydase or Cyanine-2–3 and −5 were from Jackson ImmunoResearch (RRID:AB_10015282, RRID:AB_2340612, RRID:AB_2307443, RRID:AB_2340607, RRID:AB_2340770, RRID:AB_2340826, RRID:AB_2340813, RRID:AB_2340819, RRID:AB_2339149, RRID:AB_2338512).

siRNA control (D-001810-10-05), ON-TARGETplus Mouse Rac1 (19353) siRNA - SMARTpool (L041170000005) (#2) and ON-TARGETplus Mouse PAK1 (18479) siRNA - SMARTpool (L048101000005) were from Dharmacon. mRac1 siRNA (#1) (sc-36352), mRhoA siRNA (#1) (sc-29471) and mRhoA siRNA (#2) (sc-36414) were purchased from Santa Cruz Biotechnologies.

FRAX597 and Y-27632 were purchased from Selleckchem. NSC23766 was purchased from Tocris Biosciences. FRAX1036 and G-5555 were purchased from MedChemExpress.

p27 constructs and p27 point mutants and deletion mutants in pCS2+, pGEX4T1 (Pharmacia), pET16b (Novagen), pcDNA3.1 Hygro (Invitrogen) or pQCXIP (Clontech) were described previously (***Besson et al., 2004b***; ***Serres et al., 2012***). The Myc-tagged full-length and deletion mutants of mouse Cortactin (WT, ΔABR6, ΔABR5-6, ΔABR4-6, ΔABR3-6, ΔABR, ΔABR/NTA and ΔSH3) in pME18S vector were described previously (***Katsube et al., 2004***). Full length human Cortactin was cloned by RT-PCR from IMR90 mRNA and inserted into the pcDNA3.1 Hygro Myc vector. Full-length

mouse Cortactin, Cortactin S113A and Cortactin S113D were kindly provided by Dr. Alan Mak (Queen's University, Kingston, Canada) (Webb et al., 2006) and subcloned into pQCXIP. Cortactin TA (S113A, S150A and S282A) and TD (S113D, S150D and S282D) were generated by site-directed mutagenesis from the pQCXIP-Cortactin-S113A and pQCXIP-Cortactin-S113D vectors, respectively. GFP-Tks5, kindly provided by Dr. Frederic Saltel (INSERM UMR1053, Bordeaux), was subcloned into the pQCXIP vector. pCMV6-Myc-hPAK1 was kindly provided by Dr Jonathan Chernoff (Fox Chase Cancer Center, Temple Health, Philadelphia; Addgene #12209). pGEX2TK-PAK-CD was kindly provided by Dr. John Collard (The Netherlands Cancer Institute, Amsterdam, The Netherlands). pEBB-Src-SH3-2 encoding the second SH3 domain of Nck1 fused to the myristoylation sequence of Src (myr-SH3-2) was a gift from Dr Bruce Mayer (University of Connecticut, Farmington). All plasmids used in this study were verified by sequencing.

## Cell culture and transfections

Primary MEFs were prepared as described previously (Spector, 1997; Besson et al., 2004b) from p27+/+, p27CK−/CK− or p27−/− embryos. MEFs were immortalized by infection with retroviruses encoding the human papilloma virus E6 protein and hygromycin selection. HeLa (RRID:CVCL_0030), HEK 293 (RRID:CVCL_0045), A-375 (RRID:CVCL_0132) and A549 (RRID:CVCL_0023) cells were obtained from Cell Lines Services. These cells were authenticated by short tandem repeat profiling. All cells were routinely tested to be free of mycoplasma contamination by DAPI staining.

All cells were grown at 37°C and 5% $CO_2$ in DMEM (Sigma), 4.5 g/l glucose supplemented with 10% fetal calf serum, 0.1 mM nonessential amino acids and 2 µg /ml penicillin-streptomycin. When indicated, tissue culture vessels were coated with 0.2% gelatin from bovine skin (Sigma G1393) for 1 hr. MEFs were infected with different plasmids using retroviruses produced in Phoenix ecotropic cells transfected by the calcium phosphate method. HEK293 cells were also transfected by the calcium phosphate method for 24 hr. HeLa cells were transfected using JetPrime (Polyplus transfection) according to manufacturer's instructions. For siRNA transfections in MEF E6 cells, we used Interferin (Polyplus transfection) according to manufacturer's instructions.

## Immunoprecipitation and GST pull-down

Cells were scraped and lysed in IP buffer containing 50 mM HEPES pH 7.5, 150 mM NaCl, 1 mM EDTA, 2.5 mM EGTA, 0.1% Tween20, 10% glycerol and 1% NP-40 complemented with 1 mM DTT, 10 mM $\beta$-glycerophosphate, 10 mM NaF, 10 mM sodium orthovanadate, 10 µg/ml Aprotinin, 10 µg/ml Bestatin, 10 µg/ml Leupeptin and 10 µg/ml Pepstatin A. After sonication for 10 s, cells extracts were centrifuged for 5 min at 12,500 g and supernatants were collected. Lysates (300 µg for MEF E6 and HeLa, 500 µg for HEK293) were incubated with 3 µg of indicated antibodies and 12 µl protein-A sepharose beads (IPA300, Repligen) (co-immunoprecipitation) or with recombinant GST proteins and glutathione sepharose beads (Pharmacia) (GST pull-down) at 4°C for 4 hr. Beads were then washed four times in lysis buffer and resuspended in 10 µl 4X sample buffer, boiled, and subjected to western blotting.

## Rac1-GTP pull-down assay

Pull-down assays for Rac1 activity were performed as described previously (Besson et al., 2004b; Sander et al., 1998; Malliri et al., 2000). Briefly, E6 MEFs in 100 mm plates were washed with ice-cold PBS, scraped off the plates in 400 µl cell-lysis buffer (50 mM Tris-HCl at pH 7.4, 2 mM MgCl2, 1% NP-40, 10% glycerol, 100 mM NaCl, complemented with 1 mM dithiothreitol, 10 µg/ml leupeptin, 10 µg/ml aprotinin, and 10 µg/ml pepstatin-A) on ice. Lysates were centrifuged for 5 min at 14,000 rpm at 4°C. Three hundred µl of cleared lysates were incubated for 30 min at 4°C with 8 µl of GST–PAK-CD bound to glutathione-coupled Sepharose beads (at ~1 µg GST fusion protein per µl of beads). Bead pellets were washed three times with ice-cold cell lysis buffer, resuspended in 4X sample buffer, and subjected to SDS-PAGE as described below. Ten µl of cell lysates were used for protein loading.

## Immunoblot

Cells were lysed in IP buffer as described above. Lysates and immunoprecipitates were mixed with 4X sample buffer and boiled. Proteins were resolved on 8–15% SDS-PAGE (depending on protein

size) and transferred to polyvinylidene difluoride membrane (Immobilon-P, Millipore). Membranes were blocked with PBS-T (PBS, 0.1% Tween-20), 5% non-fat dry milk and probed with indicated primary antibodies overnight at 4°C with gentle agitation. Membranes were washed three times in PBS-T then incubated with corresponding HRP-conjugated secondary antibody (1/10000) for 4 hr at room temperature. Bands were visualized using enhanced chemiluminescence detection reagents (Millipore, BioRad, Ozyme) and autoradiographic film (Blue Devil).

## Gelatin degradation assay

Coverslips were cleaned overnight in 1 M HCl, washed four times in ddH$_2$O and then coated successively with 50 µg/ml Poly L-lysine, 0.5% glutaraldehyde, fluorescent gelatin (1:10 mix of gelatin from pig skin Oregon green 488 conjugate [G13186, Molecular Probes] and 0.2% gelatin from bovine skin [Sigma G1393]), and 5 mg/ml sodium borohydride. Between each coating, coverslips were washed three times with PBS. Coverslips were then sterilized with 70% ethanol and cells were seeded and incubated either overnight or 48 hr before fixation and staining.

## Immunofluorescence

Cells were seeded on coverslips coated as described above with only non fluorescent gelatin and grown overnight. When indicated, cells were permeabilized with 20 µg/ml digitonin in PBS for 2 min before fixation. Cells were fixed with 2% PFA in PBS for 20 min at 37°C. For immunostaining, cells were permeabilized for 3 min with PBS 0.2% Triton X-100, rinsed three times in PBS and incubated for 20 min in blocking solution (PBS, 3% BSA, 0,05% Tween20 and 0,08% sodium azide) and with primary antibodies diluted in blocking solution for 1 hr. After three washes of 5 min in PBS, cells were incubated for 30 min at 37°C with Cy-2 Cy-3 or Cy-5 conjugated secondary antibodies at 1:500 dilutions. Coverslips were washed 3 times for 5 min in PBS with the first wash containing 0.1 µg/ml Hoechst H33342 and mounted on glass slides with gelvatol (20% glycerol (v/v), 10% polyvinyl alcohol (w/v), 70 mM Tris pH 8). Images were captured on a Nikon 90i Eclipse microscope using a DS-Qi2 HQ (Nikon) camera and the NIS Element software.

For live cell imaging, MEFs infected with pQCXIP-Tks5-GFP were seeded in gelatin-coated wells overnight and placed in a controlled atmosphere chamber (37°C and 5% CO$_2$) for imaging. Images were acquired every 2 min with a Zeiss Cell Observer microscope for 8 hr. Image analysis was performed with Image J/Fiji.

## Migration assays

Cells were seeded in gelatin coated 96-well plates (Essen ImageLock, Essen Bioscience) at 80% confluence and incubated overnight to allow them to reach confluence. Cells were treated for 2 hr before wounding with 2 µg/ml mitomycin C (M4287, Sigma) to block cell proliferation and scratches were performed in the cell monolayer using the wound maker (Essen Bioscience). Cells were washed immediately three times with PBS and re-fed with growth medium. Cell migration was monitored with an Incucyte FLR Live-Cell imaging system equipped with a 20X objective (Essen Bioscience). Images were acquired every 3 hr for 48 hr. Migration was quantified by measuring the relative wound density of at least three biological replicates in each experiment, using the Incucyte software (Essen Bioscience) as recommended by the manufacturer.

## Invasion assays

Fifty µl of 1 mg/ml of rat tail Collagen I (354236, Corning) solution were incubated for 1 hr at room temperature in 8 µm pores transwell inserts in 24-well plates (#353097, Corning) to allow gelling of collagen. The gel was hydrated with DMEM-50% FCS for 3 hr at 37°C. Cells were incubated for 3 hr with 2 µg/ml Mitomycin C, washed twice with DMEM 0.1% FCS, trypsinized, counted and 30,000 cells were seeded in the transwell upper chamber in 100 µl of DMEM 0.1% FCS, with 600 µl of DMEM 10% FCS in the bottom chamber. Plates were incubated for 48 hr at 37°C and 5% CO$_2$. At the end of the assay, the content of the top chamber was removed with a cotton swab and cells that invaded onto the bottom membrane of the transwell were quantified by XTT staining with 200 µg/ml XTT (Santa-Cruz Biotechnology, sc-258336) previously activated by adding 25 µM phenazine methosulfate (Sigma, P9625) in DMEM 10% FCS for 4 hr at 37°C. Absorbance was read at 450 nm.

Cells were then fixed for 5 min in 95% ethanol/5% acetic acid and stained overnight with hematoxylin to capture images of invaded cells.

## Statistical analyses

Statistical analyses were performed using Graphpad Prism 6.0 software. Differences between groups were evaluated using 1-way ANOVA followed by Bonferroni test for multiple comparison and considered significant when $p < 0.05$. Data are presented as mean ± SEM.

## Mass spectrometry analyses

HEK293 cells were transfected either with human Myc-tagged Cortactin or Myc-Cortactin and Myr-SH3-2 (that activates PAK1) for 24 hr. Cortactin was immunoprecipitated from lysates from four 100 mm plates per IP using 12 µg per IP of Mouse anti-Myc 9E10 antibody. Samples were separated on 12% SDS-PAGE. Gel was stained with instant blue for 1 hr. Amounts of immunoprecipitated Cortactin was estimated to 4 ug for each IP by comparison with a BSA standard on the same gel. Bands of interest were cut and placed in eppendorf tubes in 30 µl of 1% acetic acid solution and stored at −20C until MS/MS analysis.

### Protein digestion

Excised gel bands were treated to remove salts, buffers and detergent using 3 washes of 50 mM $NH_4HCO_3$ intercalated with gel shrinkages in 50% acetonitrile (ACN). Disulfide bonds were disrupted with 20 mM DTT at 56°C for 30 min and subsequently alkylated in 55 mM chloroacetamide for 30 min at room temperature. A last wash with 50% ACN and 50 mM $NH_4HCO_3$ was performed before addition of 50 ng of trypsin (Promega Sequencing Grade) for an overnight digestion on an Eppendorf Thermomix at 30°C. Peptides were then extracted using 2 washes of 1% formic acid intercalated with gel shrinkages in 50% ACN. The two washes were pooled and evaporated before analysis.

### LC-MS/MS

20% of each sample was analyzed in LC-MS-MS using an Ultimate 3000 Rapid Separation liquid chromatographic system coupled to an Orbitrap Fusion mass spectrometer (both from Thermo Fisher Scientific). Peptides were directly loaded on a $C_{18}$ reverse phase analytical column (2 µm particle size, 100 Å pore size, 75 µm internal diameter, 50 cm length) with a 45 min effective gradient from 99% A (0.1% formic acid and 100% H2O) to 50% B (80% ACN, 0.085% formic acid and 20% H2O). The Orbitrap Fusion mass spectrometer acquired data throughout the elution process and operated in a data dependent scheme with full MS scans acquired with the orbitrap, followed by stepped HCD MS/MS (top speed mode in 3 s) on the most abundant ions detected in the MS scan. Mass spectrometer settings were: full MS (AGC: $4.0E^5$, resolution: 120,000, m/z range 350–1500, maximum ion injection time: 60 ms); MS/MS was performed using Stepped HCD Normalized Collision Energy at 30 plus and minus 5%, Orbitrap resolution was set at 17500, intensity threshold: $1.0E^4$, isolation window: 1.6 m/z, dynamic exclusion time setting: 30 s, AGC Target: $1.0E^4$ and maximum injection time: 100 ms). The fragmentation was permitted of precursor with a charge state of 2 to 4. The software used to generate. mgf files is Proteome discoverer 1.4.

### Peptide identification

Mass list were used to perform comparison of MS/MS experimental peak lists with the *Homo sapiens* taxon of the Swiss-Prot database (November 2016, 20,204 sequences) using Mascot version 2.5.1 (*Perkins et al., 1999*). The cleavage specificity set was the trypsin with maximum four missed cleavages. The precursor mass tolerance was set to 4 ppm and the MS/MS mass tolerance to 0.55 Da. Cystein carbamidomethylation was set as a constant modification while methionine oxidation, tyrosine, serine and threonine phosphorylation were set as variable modification. With these settings, peptide identifications were considered as valid whenever their scores reached a minimum of 23, thus meeting the *p*-values criteria less than 0.05.

## Acknowledgements

We are very grateful to Dr Alan Mak (Queen's University, Kingston, Canada), Dr Jonathan Chernoff (Fox Chase Cancer Center, Philadelphia, USA), Dr Bruce Mayer (University of Connecticut, Farmington) and Dr John Collard (The Netherlands Cancer Institute, Amsterdam, the Netherlands) for providing reagents. We thank Dr Helène Chanut-Delalande (CNRS UMR5547, Toulouse) and Dr Violaine Moreau and Dr Frédéric Saltel (INSERM UMR1053, Bordeaux) for stimulating discussions and reagents. We thank Emilie-Fleur Gautier, Virginie Salnot, Cedric Broussard, Marjorie Leduc (3P5 proteomics facility of the Université Paris Descartes), Thomas Daunizeau, Dr Christine Jean (CRCT UMR1037, Toulouse) and Laetitia Ligat (Cell imaging facility, CRCT) for technical assistance. P. J. was supported by a studentship from the Ministère de l'Enseignement Supérieur et de la Recherche. A.N. is supported by a studentship from the Ministère de l'Enseignement Supérieur et de la Recherche. R.P. is supported by a studentship from the Ligue Nationale Contre le Cancer. The Orbitrap Fusion mass spectrometer was acquired with funds from the FEDER through the "Operational Programme for Competitiveness and Employment 2007-2013" and from the "Canceropole Ile de France". S.M. is supported by funds from the Ligue Nationale Contre le Cancer. A.B. is supported by grants from the Fondation ARC pour la Recherche sur le Cancer and Ligue Nationale Contre le Cancer.

## Additional information

### Funding

| Funder | Author |
| --- | --- |
| Ligue Nationale Contre le Cancer | Renaud T Perchey<br>Stéphane Manenti<br>Arnaud Besson |
| Ministère de l'Enseignement Supérieur et de la Recherche Scientifique | Pauline Jeannot<br>Ada Nowosad |
| Institut National de la Santé et de la Recherche Médicale | Evangeline Bennana<br>Patrick Mayeux<br>François Guillonneau<br>Stéphane Manenti<br>Arnaud Besson |
| Centre National de la Recherche Scientifique | Stéphane Manenti<br>Arnaud Besson |
| Canceropole Ile de France | François Guillonneau<br>Patrick Mayeux |

The funders had no role in study design, data collection and interpretation, or the decision to submit the work for publication.

### Author contributions

PJ, Conceptualization, Data curation, Formal analysis, Validation, Investigation, Methodology, Writing—original draft; AN, Conceptualization, Data curation, Investigation, Methodology; RTP, Data curation, Investigation, Methodology; CC, Investigation, Methodology; EB, Data curation, Formal analysis, Investigation, Methodology; TK, Resources, Methodology; PM, Conceptualization, Data curation, Formal analysis, Investigation; FG, Conceptualization, Data curation, Formal analysis, Investigation, Methodology, Writing—review and editing; SM, Funding acquisition, Writing—original draft, Project administration; AB, Conceptualization, Resources, Data curation, Formal analysis, Supervision, Funding acquisition, Validation, Investigation, Methodology, Writing—original draft, Project administration, Writing—review and editing

### Author ORCIDs

Arnaud Besson, http://orcid.org/0000-0002-9599-3943

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
