## [Decision Letter]

[Editors’ note: a previous version of this study was rejected after peer review, but the authors submitted for reconsideration. The first decision letter after peer review is shown below.]

Thank you for submitting your work entitled "p27^Kip1^ promotes invadopodia turnover and invasion through the regulation of the PAK1/Cortactin pathway" for consideration by *eLife*. Your article has been reviewed by three peer reviewers, one of whom is a member of our Board of Reviewing Editors, and the evaluation has been overseen by Fiona Watt as the Senior Editor.

Our decision has been reached after consultation between the reviewers. Based on these discussions and the individual reviews below, we regret to inform you that your work will not be considered further for publication in *eLife*.

This study demonstrates that p27 binding to Cortactin promotes PAK1 interactions with Cortactin, and invadopodia disassembly by phosphorylation of Cortactin on Ser-113. Biochemical analysis is presented to support these conclusions, including the demonstration that a phosphomimetic mutant of Cortactin can complement the p27 deficiency phenotype on invadopodia disassembly. Together, these aspects of the manuscript represent significant strengths. However, the detailed reviews (appended) identify a number of weaknesses in the data used to support the presented conclusions. Given the extensive new experimentation that will be required to address these weaknesses, a decision to reject your manuscript was made. Please be assured that *eLife* is interested in this area of research. Indeed, we would welcome the submission of a new manuscript on this topic when findings from new experimentation are obtained.

Reviewer #1:

This study demonstrates that p27 binding to Cortactin promotes PAK1 interactions with Cortactin, and invadopodia disassembly by phosphorylation of Cortactin on Ser-113. In general, the biochemical analysis is strong and convincing. Indeed, the demonstration that a phosphomimetic mutant of Cortactin can complement the p27 deficiency phenotype on invadopodia disassembly is very interesting. However, there are a number of areas that require strengthening.

1) The siRNA studies (Rac and RhoA) were performed with a single oligonucleotide. To exclude off0target effects, phenotypes should be confirmed with two independent siRNAs.

2) Some care should be taken concerning the PAK1/Cortactin connection. First, the studies presented are focussed on mutational analysis of the Cortactin phosphorylation site Ser-113 rather than the kinase. Second, in vitro studies demonstrate phosphorylation of Cortactin by PAK3 at more than one site (Webb et al.). Third, what is the evidence that PAK1 phosphorylates Cortactin in vivo? Fourth, the in vivo studies are reliant on a drug.

3) The conclusion that p27 regulates Cortactin/PAK mediated invadopodia disassembly are based on static images of invadopodia number. Conclusions about the dynamic nature of invadopodia would be best supported by images documenting the dynamics.

Reviewer #2:

This work, building on prior publications from Webb et al. and Moshfegh et al., 2014 attempts to provide a mechanistic basis for the effects of cytosolic p27 on invadopodia dynamics and cell invasion. For the most part, the data appear sound, but there are a few key issues that are not adequately addressed.

1) The only experiment that identifies Pak1 as a link between p27 and invadopodia is treatment of MEFs with FRAX597 (Figure 6). There are two issues here: this compound affects all three group A Paks (thus one can't say the effects are through Pak1) and FRAX597 has poor selectivity towards STE20 kinases such as Group A Paks as compared to tyrosine kinases. There are much more selective compounds (e.g., FRAX1036 and G5555) as well as other approaches (siRNA, CRISPR) that could and should be used to establish the link claimed by the authors. As things stand, the data do not adequately support the author's contentions.

2) There really isn't any proof that Cortactin gets phosphorylated by Pak at S113, only that substitutions at this residue affect invadopodia dynamics (Figure 7). What would have made the case much stronger are phospho-S113 blots and/or IFs showing that phosphorylation of this site is modulated by Rac/Pak. As far as I can tell, the initial paper from Webb et al., 2006, on which the present manuscript relies, only provided in vitro evidence that recombinant Pak3 could phosphorylate Cortactin at S113. As there are no available phospho S113 antibodies, the authors might instead IP Cortactin from p27-/- MEFs and blot with p-Ser antibodies or, better, ID sites by mass spec. I know this sort of experiments isn't fun or easy, but the use of phospho-mimics alone is not adequate to support their conclusions.

Reviewer #3:

This manuscript identifies p27^kip1^ as a regulator of invadopodia formation and cell invasion, increasing the surprising number of cellular processes known to be regulated by this protein. While cytoplasmic p27^kip1^ was previously established to limit RhoA activity and stress fiber formation to promote cell migration, Jeannot et al., 2015 describe a new mechanism, where p27 promotes Cortactin phosphorylation by Pak1 to reduce invadopodia and increase cell invasion through collagen. While the authors include a strong set of experiments that convincingly demonstrates a p27/Cortactin/PAK1 pathway exists, the manuscript overall feels somewhat preliminary because it is not clear how inhibition of this pathway can increase invadopodia formation and prevent cell invasion. Presenting a mechanism to explain this puzzling observation and more clearly demonstrating p27 is localized to matrix degrading structures would strengthen the novel, and important conclusions made by the authors. Additional concerns are listed below.

Major Concerns:

The pre-fixation permeabilization/extraction of the cytoplasm prior to staining for p27 does not give a lot of confidence that p27 is localized to invadopodia (as in Figure 2, for example). Additionally, the Cortactin phosphomimetic construct decreases the number of invadopodia without significantly reducing the area of gelatin degradation, raising the strong possibility that the structures being identified by Cortactin (eg. Figure 3) are not actual invadopodia, despite their punctate appearance. To make confirm these structures are invadopodia, the authors should show p27 is enriched in Cortactin or Tsk5 positive puncta which coincide with regions of matrix degradation.

The authors' measure number of cells with invadopodia (eg. Figure 3), yet often conclude that cells have more invadopodia. If the authors are going to conclude that there are more invadopodia per cell, this is something that must be explicitly measured.

Importantly, the strikingly counterintuitive conclusion that increasing invadopodia inhibits invasion was not sufficiently justified. It remains a possibility that another pathway mediated by p27 is responsible for the invasion phenotype. One way to address this possibility would be to show expression of p27 1-190 is unable to rescue cell invasion.

It is not clear if the migration defect resulting from loss of p27Kip (Besson et al., 2006), is mechanistically distinct from the invasion phenotype reported here. In their previous publication, it was shown that inhibition of Rho kinase rescued migration. Does Rho kinase inhibition also restore the ability of the p27 -/- cells to invade through thin collagen layers? If not, it would strengthen the conclusion that the authors are investigating a novel migration mechanism downstream of p27^kip1^.

[Editors’ note: what now follows is the decision letter after the authors submitted for further consideration.]

Thank you for resubmitting your work entitled "p27^Kip1^ promotes invadopodia turnover and invasion through the regulation of the PAK1/Cortactin pathway" for further consideration at *eLife*. Your revised article has been favorably evaluated by Fiona Watt (Senior editor) and three reviewers, one of whom is a member of our Board of Reviewing Editors.

The manuscript has been improved but there are some remaining issues that need to be addressed before acceptance, as outlined below:

1) While the colocalization of p27 and Tks5 in regions of degraded gelatin is now included in the manuscript, increasing confidence that these structures are bona fide invadopodia, it is not clear how specific the colocalization is compared to regions of the cell that are not associated with regions of gelatin degradation. I suggest the authors include line scans of regions of the cell not associated with gelatin degradation to demonstrate the specificity and import of the colocalization they now measure near regions of degraded gelatin.

2) It is not clear if statistical tests were performed on the data presented in Figure 2, Figure 2—figure supplement 2, Figure 5, and Figure 8—figure supplement 1.

[Editors' note: further revisions were requested prior to acceptance, as described below.]

Thank you for resubmitting your work entitled "p27^Kip1^ promotes invadopodia turnover and invasion through the regulation of the PAK1/Cortactin pathway" for further consideration at *eLife*. Your revised article has been favorably evaluated by Fiona Watt (Senior editor) and a Reviewing editor.

The manuscript has been improved but there is one minor issue that needs to be addressed before acceptance. The new statistical analysis of Figure 2 and Figure 2—figure supplement 2 stated in the figure legends indicates that differences are not significant. However, in the text referring to these figures, it is stated that increases were observed. This is contradictory. Please clarify.

---

## [Author Response]

[Editors’ note: the author responses to the first round of peer review follow.]

Reviewer #1:

This study demonstrates that p27 binding to Cortactin promotes PAK1 interactions with Cortactin, and invadopodia disassembly by phosphorylation of Cortactin on Ser-113. In general, the biochemical analysis is strong and convincing. Indeed, the demonstration that a phosphomimetic mutant of Cortactin can complement the p27 deficiency phenotype on invadopodia disassembly is very interesting. However, there are a number of areas that require strengthening.

1) The siRNA studies (Rac and RhoA) were performed with a single oligonucleotide. To exclude off0target effects, phenotypes should be confirmed with two independent siRNAs.

To strengthen the data obtained with single siRNAs for Rac1 and RhoA, we have repeated these experiments (# of cells forming invadopodia and area degraded/cell) with a pool of 4 siRNA against Rac1 (ON-TARGETplus Mouse Rac1 siRNA – SMARTpool (L041170000005)) and with a new pool of 3 siRNAs against RhoA (sc-36414). These results have been incorporated into Figure 7 for Rac1 and Figure 7—figure supplement 1 for RhoA. In addition, siRac1 results are also replicated using a pharmacological inhibitor of Rac1 (Figure 7), arguing against an off-target effect of the siRNA.

*2) Some care should be taken concerning the PAK1/Cortactin connection. First, the studies presented are focussed on mutational analysis of the Cortactin phosphorylation site Ser-113 rather than the kinase. Second,* in vitro *studies demonstrate phosphorylation of Cortactin by PAK3 at more than one site (Webb et al.). Third, what is the evidence that PAK1 phosphorylates Cortactin* in vivo*? Fourth, the* in vivo *studies are reliant on a drug.*

We have modified the text and use more careful wording when describing the PAK1/Cortactin connection. We also have taken several steps to strengthen this point:

1) To investigate the importance of PAK1, we had data using the FRAX597 inhibitor, which targets the ATP-binding pocket in the kinase domain of PAK1-3 and inhibits the kinase activity of PAK, therefore suggesting that the kinase activity of PAK is involved.

As suggested by reviewer 2, we replicated these experiments (# of cells forming invadopodia and area degraded/cell) and with other, more specific inhibitors, of PAK1-3 (FRAX1036 and G5555). These inhibitors gave similar results as FRAX597, thus confirming the importance of PAK activity in p27-mediated control of invadopodia dynamics. This data has been included in the manuscript as Figure 6—figure supplement 1.

Since these inhibitors cannot discriminate between PAK1-3, we also performed similar experiments with siRNAs (a pool of 4 siRNAs) against PAK1 (ON-TARGETplus Mouse PAK1 (18479) siRNA – SMARTpool (L048101000005). Results with PAK1 siRNA are similar to those obtained with PAK pharmacological inhibitors, supporting the involvement of PAK1 in the pathway we describe. These data have been included in Figure 6.

Altogether, we now have data from 3 different PAK1-3 pharmacological inhibitors and from PAK1-specific siRNA that give similar results, thus strongly suggesting that PAK1 is indeed involved in the pathway regulating invadopodia dynamics.

2) Webb and colleagues (2006) have reported the phosphorylation of Cortactin by PAK3 in vitro on three different residues (S113, S150 and S282).

As suggested by the reviewer, we tested whether the other sites known to be phosphorylated by PAK on Cortactin were important in the phenotype observed, We generated triple S113/S150/S282 unphosphorylatable and phosphomimetic Cortactin mutants and measured the effect on # of cells forming invadopodia and area degraded/cell. Triple mutants give similar results as S113 Cortactin mutants, with a stronger effect of the triple unphosphorylatable mutant in p27+/+ cells (13.5% cells forming invadopodia with S113A versus 23.3% with tripleA mutant; 6.17 fold increased area degraded with S113A versus 10.02 fold with TripleA) suggesting that these sites (or at least one of them) also contribute to Cortactin regulation of invadopodia dynamics. These results have been included in Figure 7. These results also reinforce the idea that phosphorylation of Cortactin is important in the control of invadopodia dynamics.

3) We now provide 4 lines of evidence indicating that these sites are indeed phosphorylated in vivo:

a) By IP Cortactin and anti Phospho-Ser immunoblot we found that Ser phosphorylation of Cortactin was reduced in presence of PAK1-3 inhibitor FRAX597. This data is provided as Figure 8—figure supplement 1.

b) By IP Cortactin and anti Phospho-Ser immunoblot we found that expression of Cortactin S113A and Cortactin S113A/S150A/S282A exhibit decreased phospho-Ser signal compared to wild-type Cortactin, arguing that phosphorylation of these sites contribute to the phospho-Ser signal detected. This data is provided as Figure 8—figure supplement 1.

c) Our initial attempt to obtain mass spectrometry data showing that Cortactin was phosphorylated in vivo was with a private company. We since established a collaboration with a group specialized in MS work at the Institut Cochin. We were able to detect a phosphorylation of Cortactin at Ser150 (one of the 3 sites reported to be phosphorylated by PAK1-3) when a myristoylated form of the second SH3 domain of Nck1 was co-expressed (previously shown to activate PAK1 by Bruce Mayer's lab: Lu et al., 1997 and Lu and Mayer, 1999) but not when only Cortactin was expressed. Therefore PAK activation in these cells appear to induce Cortactin S150 phosphorylation. This mass spectrometry data is provided as Figure 8—figure supplement 2.

d) Database mining (PhosphositePlus) shows that Cortactin phosphorylation on S113, S150 and/or S282, the three sites targeted by PAK1-3, has previously been detected in vivo by mass spectrometry and this is described in 8 publications of global proteome analyses:

1) Chen et al., J Proteome Res 2010, 9:174-8

Shows Cortactin S113 phosphorylation in LNCaP prostate cancer cells

2)Olsen et al., Sci Signal 2010, 3(104):ra3

Shows Cortactin S282 phosphorylation in mitotic cells

3) Rinschen et al., PNAS 2010, 107: 3882-7

Shows Cortactin S150 phosphorylation in mpkCCD renal cells

4) Wisniewski et al., J Proteome Res 2010, 9:3280-9

Shows Cortactin S113 phosphorylation in mouse brain

5) Klammer et al., Mol Cell Proteomics 2012, 11: 651-88

Shows Cortactin S282 phosphorylation in breast tumors

6) Bian et al., J Proteomics 2014, 96: 253-62

Shows Cortactin S150 phosphorylation in human liver

7) Mertins et al., Mol Cell Proteomics 13: 1690-704

Show Cortactin S113, S150 and S282 phosphorylation in ovarian and breast xenograft tissues

8) Mertins et al., Nature 2016, 534: 55-62

Shows Cortactin S113 and S282 phosphorylation in breast cancer

These references are now discussed in our revised manuscript.

We believe that the fact several other groups have reported the phosphorylation of those sites in vivo in various models in independent studies provides strong evidence that Cortactin is actually phosphorylated on these three sites in vivo.

4) As detailed in point #1, our in vivo studies are now relying on three different PAK1-3 pharmacological inhibitor as well as similar experiments performed with PAK1 siRNA treated cells.

3) The conclusion that p27 regulates Cortactin/PAK mediated invadopodia disassembly are based on static images of invadopodia number. Conclusions about the dynamic nature of invadopodia would be best supported by images documenting the dynamics.

To strengthen our conclusions and provide direct evidence that p27 affects invadopodia stability, we determined invadopodia lifetime in p27+/+ and p27-/- MEFS by videomicroscopy using eGFP-Tks5. This data has been added as Figure 5 and shows that invadopodia in p27-/- cells have a longer lifetime than in p27+/+ cells.

Reviewer #2:

This work, building on prior publications from Webb et al. and Moshfegh et al., 2014 attempts to provide a mechanistic basis for the effects of cytosolic p27 on invadopodia dynamics and cell invasion. For the most part, the data appear sound, but there are a few key issues that are not adequately addressed.

1) The only experiment that identifies Pak1 as a link between p27 and invadopodia is treatment of MEFs with FRAX597 (Figure 6). There are two issues here: this compound affects all three group A Paks (thus one can't say the effects are through Pak1) and FRAX597 has poor selectivity towards STE20 kinases such as Group A Paks as compared to tyrosine kinases. There are much more selective compounds (e.g., FRAX1036 and G5555) as well as other approaches (siRNA, CRISPR) that could and should be used to establish the link claimed by the authors. As things stand, the data do not adequately support the author's contentions.

As suggested by the reviewer, we replicated these experiments (# of cells forming invadopodia and area degraded/cell) with siRNAs against PAK1 and with the more specific inhibitors of PAK1-3, FRAX1036 and G5555, to better support the link between PAK1 and Cortactin in p27-mediated regulation of invadopodia dynamics. These results are provided in Figure 6 and in Figure 6—figure supplement 1.

*2) There really isn't any proof that Cortactin gets phosphorylated by Pak at S113, only that substitutions at this residue affect invadopodia dynamics (Figure 7). What would have made the case much stronger are phospho-S113 blots and/or IFs showing that phosphorylation of this site is modulated by Rac/Pak. As far as I can tell, the initial paper from Webb et al., 2006, on which the present manuscript relies, only provided* in vitro *evidence that recombinant Pak3 could phosphorylate Cortactin at S113. As there are no available phospho S113 antibodies, the authors might instead IP Cortactin from p27-/- MEFs and blot with p-Ser antibodies or, better, ID sites by mass spec. I know this sort of experiments isn't fun or easy, but the use of phospho-mimics alone is not adequate to support their conclusions.*

As suggested by the reviewer, we performed additional experiments indicating that:

a) By IP Cortactin and anti Phospho-Ser immunoblot we found that Ser phosphorylation of Cortactin was reduced in presence of PAK1-3 inhibitor FRAX597. This suggests that PAK1-3 contribute to the phosphorylation of Cortactin. This data is provided as Figure 8—figure supplement 1.

b) By IP Cortactin and anti Phospho-Ser immunoblot we found that expression of Cortactin S113A and Cortactin S113A/S150A/S282A (the sites targeted by PAK) exhibit decreased phospho-Ser signal compared to wild-type Cortactin, arguing that phosphorylation of these sites contribute to the phospho-Ser signal detected. This data is provided as Figure 8—figure supplement 1.

c) Our initial attempt to obtain mass spectrometry data showing that Cortactin was phosphorylated in vivo was with a private company. We since established a collaboration with a group specialized in MS work at the Institut Cochin. We were able to detect a phosphorylation of Cortactin at Ser150 (one of the 3 sites reported to be phosphorylated by PAK1-3) when a myristoylated form of the second SH3 domain of Nck1 was co-expressed (previously shown to activate PAK1 by Bruce Mayer's lab: Lu et al., 1997 and Lu and Mayer, 1999) but not when only Cortactin was expressed. Therefore PAK activation in these cells appear to induce Cortactin S150 phosphorylation. This mass spectrometry data is provided as Figure 8—figure supplement 2.

d) Database mining (PhosphositePlus) shows that Cortactin phosphorylation on S113, S150 and/or S282, the three sites targeted by PAK1-3, has previously been detected in vivo by mass spectrometry and this is described in 8 publications of global proteome analyses:

1) Chen et al., J Proteome Res 2010, 9:174-8

Shows Cortactin S113 phosphorylation in LNCaP prostate cancer cells

2) Olsen et al., Sci Signal 2010, 3(104):ra3

Shows Cortactin S282 phosphorylation in mitotic cells

3) Rinschen et al., PNAS 2010, 107: 3882-7

Shows Cortactin S150 phosphorylation in mpkCCD renal cells

4) Wisniewski et al., J Proteome Res 2010, 9:3280-9

Shows Cortactin S113 phosphorylation in mouse brain

5) Klammer et al., Mol Cell Proteomics 2012, 11: 651-88

Shows Cortactin S282 phosphorylation in breast tumors

6) Bian et al., J Proteomics 2014, 96: 253-62

Shows Cortactin S150 phosphorylation in human liver

7) Mertins et al., Mol Cell Proteomics 13: 1690-704

Show Cortactin S113, S150 and S282 phosphorylation in ovarian and breast xenograft tissues

8) Mertins et al., Nature 2016, 534: 55-62

Shows Cortactin S113 and S282 phosphorylation in breast cancer

These references are now discussed in our revised manuscript.

Reviewer #3:

This manuscript identifies p27^kip1^ as a regulator of invadopodia formation and cell invasion, increasing the surprising number of cellular processes known to be regulated by this protein. While cytoplasmic p27^kip1^ was previously established to limit RhoA activity and stress fiber formation to promote cell migration, Jeannot et al, 2015. describe a new mechanism, where p27 promotes Cortactin phosphorylation by Pak1 to reduce invadopodia and increase cell invasion through collagen. While the authors include a strong set of experiments that convincingly demonstrates a p27/Cortactin/PAK1 pathway exists, the manuscript overall feels somewhat preliminary because it is not clear how inhibition of this pathway can increase invadopodia formation and prevent cell invasion. Presenting a mechanism to explain this puzzling observation and more clearly demonstrating p27 is localized to matrix degrading structures would strengthen the novel, and important conclusions made by the authors. Additional concerns are listed below.

Major Concerns:

The pre-fixation permeabilization/extraction of the cytoplasm prior to staining for p27 does not give a lot of confidence that p27 is localized to invadopodia (as in Figure 2, for example). Additionally, the Cortactin phosphomimetic construct decreases the number of invadopodia without significantly reducing the area of gelatin degradation, raising the strong possibility that the structures being identified by Cortactin (eg. Figure 3) are not actual invadopodia, despite their punctate appearance. To make confirm these structures are invadopodia, the authors should show p27 is enriched in Cortactin or Tsk5 positive puncta which coincide with regions of matrix degradation.

We performed co-immunostaining for Tks5, p27 and fluorescent gelatin providing evidence that p27 and Tks5 colocalize in structures capable of degrading ECM. These data in p27+/+ MEFs and in A549 lung adenocarcinoma cells are now presented in Figure 3—figure supplement 2.

We had some variability in the last set of experiments using S113 Cortactin mutants. More experiments have now been performed and these experiments have been incorporated in the existing figure panels, improving the quality of the data presented (Figure 8). In addition, we now provide new data using Cortactin mutants in which the three sites reported to be phosphorylated by PAK (S113/S150/S282) have been mutated to either unphosphorylatable and phosphomimetic residues. The triple unphosphorylatable mutant (Cortactin TA) give more robust responses in p27+/+ cells (13.5% cells forming invadopodia with S113A versus 23.3% with tripleA mutant; 6.17 fold increased area degraded with S113A versus 10.02 fold with TripleA) suggesting that these sites have a synergistic effect in Cortactin regulation of invadopodia dynamics. This data is presented in Figure 8.

The authors' measure number of cells with invadopodia (eg. Figure 3), yet often conclude that cells have more invadopodia. If the authors are going to conclude that there are more invadopodia per cell, this is something that must be explicitly measured.

We have modified the text to ensure that our conclusions are referring only to the number of cells forming invadopodia.

Importantly, the strikingly counterintuitive conclusion that increasing invadopodia inhibits invasion was not sufficiently justified. It remains a possibility that another pathway mediated by p27 is responsible for the invasion phenotype. One way to address this possibility would be to show expression of p27 1-190 is unable to rescue cell invasion.

As suggested by the reviewer, we performed invasion assays with p27-/- MEFs infected with either empty vector, p27CK- or p27CK- 1-190 mutants. While p27CK- expression clearly increased the invasive capacity of p27-/- cells, the p27CK- 1-190 mutant had no effect, thus confirming that the ability of p27 to interact with Cortactin is required for the regulation of invasion. This data is now presented in Figure 4.

It is not clear if the migration defect resulting from loss of p27Kip (Besson et al., 2006), is mechanistically distinct from the invasion phenotype reported here. In their previous publication, it was shown that inhibition of Rho kinase rescued migration. Does Rho kinase inhibition also restore the ability of the p27 -/- cells to invade through thin collagen layers? If not, it would strengthen the conclusion that the authors are investigating a novel migration mechanism downstream of p27^kip1^.

As the reviewer suggested, we performed transwell invasion assays with cells treated with the ROCK inhibitor Y26632. ROCK inhibition had no effect on the invasive capacity of p27+/+ and p27-/- MEFs (Figure 7—figure supplement 1). This is in agreement with our data indicating that siRNAs against RhoA did not significantly affect the number of cells forming invadopodia and only slightly increased the ability of both p27+/+ and p27-/- MEFs to degrade ECM (Figure 7—figure supplement 1), suggesting that the regulation of RhoA by p27 is not involved in this phenotype.

[Editors' note: the author responses to the re-review follow.]

The manuscript has been improved but there are some remaining issues that need to be addressed before acceptance, as outlined below:

1) While the colocalization of p27 and Tks5 in regions of degraded gelatin is now included in the manuscript, increasing confidence that these structures are bona fide invadopodia, it is not clear how specific the colocalization is compared to regions of the cell that are not associated with regions of gelatin degradation. I suggest the authors include line scans of regions of the cell not associated with gelatin degradation to demonstrate the specificity and import of the colocalization they now measure near regions of degraded gelatin.

As requested, we added line scans on the same images not associated with regions of gelatin degradation and the results indicate that individual spots of Tks5 and p27 do not colocalize extensively outside of these regions of gelatin degradation. Two panels (C and D) have been added to Figure 3—figure supplement 2 to illustrate this point.

2) It is not clear if statistical tests were performed on the data presented in Figure 2, Figure 2—figure supplement 2, Figure 5, and Figure 8—figure supplement 1.

Statistical analyses had not been performed on the graphs. We have now performed these analyses:

For Figure 2, Figure 2—figure supplement 2 and Figure 5, the differences of each time-point with the control condition (time 0) were not significant. We added a sentence in the figure legends of these panels to mention that fact. For the graphs in Figure 5 and Figure 8—figure supplement 1, the differences were statistically significant (p<0.01) and we added ‘**’ on the graphs and legends. All statistical analyses for these panels are now included in the source data files accompanying the manuscript.

[Editors' note: further revisions were requested prior to acceptance, as described below.]

The manuscript has been improved but there is one minor issue that needs to be addressed before acceptance. The new statistical analysis of Figure 2 and Figure 2—figure supplement 2 stated in the figure legends indicates that differences are not significant. However, in the text referring to these figures, it is stated that increases were observed. This is contradictory. Please clarify.

Original sentence:

“Consistent with the translocation of p27 in the cytoplasm and the formation of invadopodia after serum or growth factor stimulation, an increased amount of Cortactin co-immunoprecipitated with p27 at 1 h and 3 h post-stimulation in p27+/+ E6 MEFs (Figure 2) and in Hela cells (Figure 2—figure supplement 2).”

New sentence:

“Consistent with the translocation of p27 in the cytoplasm and the formation of invadopodia after serum or growth factor stimulation, an increased association between p27 and Cortactin by co-immunoprecipitation at 1 h and 3 h post-stimulation in p27+/+ E6 MEFs (Figure 2) and in Hela cells (Figure 2—figure supplement 2) was observed in five and three independent experiments, respectively. However, due to variability in signal intensities among independent experiments, these differences were not statistically significant.”